

# Optical flow gas velocity analysis in plumes using UV cameras – Implications for SO$_2$-emission-rate retrievals investigated at Mt. Etna, Italy, and Guallatiri, Chile

Jonas Gliß[1,2,3], Kerstin Stebel[1], Arve Kylling[1], and Aasmund Sudbø[3]

[1]Norwegian Institute for Air Research, Kjeller, Norway
[2]Department of Physics, University of Oslo (UiO), Oslo, Norway
[3]Department of Technology Systems, University of Oslo (UiO), Kjeller, Norway

*Correspondence to:* Jonas Gliß (jg@nilu.no)

**Abstract.** Accurate gas velocity measurements in emission plumes are highly desirable for various atmospheric remote sensing applications. The imaging technique of UV SO$_2$ cameras is commonly used for monitoring of SO$_2$ emissions from volcanoes and anthropogenic sources (e.g. power plants, ships). The camera systems capture the emission plumes at high spatial and temporal resolution. This allows to retrieve the gas velocities in the plume directly from the images. The latter can be mea-

sured at a pixel level using optical flow (OF) algorithms. This is particularly advantageous under turbulent plume conditions. However, OF algorithms intrinsically rely on contrast in the images and often fail to detect motion in low-contrast image areas. We present a new method to identify ill-constraint OF motion-vectors and replace them using the local average velocity vector. The latter is derived based on histograms of the retrieved OF motion-fields. The new method is applied to two example datasets recorded at Mt. Etna (Italy) and Guallatiri (Chile). We show that in many cases, the uncorrected OF yields significantly un-

derestimated SO$_2$-emission-rates. We further show, that our proposed correction can account for this and that it significantly improves the reliability of optical flow based gas velocity retrievals.

In the case of Mt. Etna, the SO$_2$ emissions of the north-east crater are investigated. The corrected SO$_2$-emission-rates range between $4.8 - 10.7\,\mathrm{kg/s}$ (average: $7.1 \pm 1.3\,\mathrm{kg/s}$) and are in good agreement with previously reported values. For the Guallatiri data, the emissions of the central crater and a fumarolic field are investigated. The retrieved SO$_2$-emission-rates are

between $0.5 - 2.9\,\mathrm{kg/s}$ (average: $1.3 \pm 0.5\,\mathrm{kg/s}$) and provide the first report of SO$_2$ emissions from this remotely located and inaccessible volcano.

## 1   Introduction

Studying and monitoring of gas emissions is highly desirable since the emitted gases can have substantial environmental impacts. This includes both natural and anthropogenic sources, for instance, volcanoes, industrial areas, power plants, urban

emissions or wildfires. The measurements can help to better assess regional and global impacts of the emissions, for instance related to air-quality standards and pollution monitoring or climate impacts (e.g. Schwartz, 1994, Robock, 2000, IPCC, 2013). Sulfur dioxide (SO$_2$), in particular, is a toxic gas emitted both by anthropogenic and natural sources, (e.g. power plants, ships,



volcanoes). The pollutant has various impacts, both of socio-environmental and economic nature (e.g. human health, agriculture) and on the climate (e.g. being a precursor of stratospheric sulfur aerosols, Wigley, 1989). It is furthermore an important monitoring parameter related to volcanic risk assessment (e.g. Fischer et al., 1994, Caltabiano et al., 1994).

Passive remote sensing techniques are commonly used for monitoring of gas emissions from localised emitters (or point sources). The instruments are based on the principle of light absorption and typically measure path integrated concentrations (column densities, CDs) of the gases. Instrumentation can be ground, airborne and satellite based and can cover wavelengths ranging from the near ultraviolet (UV) up to thermal long-wave infrared (LWIR), either using solar or thermal radiation as light source. Note that due to the variety of different technologies, the term "point-source" is not clearly defined and may, in some
cases, refer to scales of several kilometres (e.g. a whole city in case of space based observations), and in other cases, to only a few metres (e.g. a power-plant chimney for ground based near-source measurements).
Gas-emission-rates (or fluxes) of the sources are typically retrieved along a plume transect downwind the source. This is done by multiplying the measured CDs with the gas velocities in the plume. The latter may be estimated using meteorological weather data (e.g. Frins et al., 2011) or using correlation techniques (if multiple scanners are available, e.g. Williams-Jones
et al., 2006) in case the measurements are performed at moderate sampling rate (e.g. spectroscopic instrumentation such as COSPEC or scanning DOAS instruments, e.g. Moffat and Millan, 1971, Platt and Stutz, 2008) and at sufficient source distance.

Another type of available instrumentation is based on camera systems which are equipped with wavelength selective filters (e.g. Mori and Burton, 2006, Prata and Bernardo, 2014, Kuhn et al., 2014, Dekemper et al., 2016). The imaging devices can
be used to create instantaneous CD maps of the measured species (e.g. $SO_2$, $NO_2$) at high spatial resolution and at sampling rates potentially down into the sub-Hz regime (dependent on the optical setup and lighting conditions). This allows to study high frequency variations in the emission signals or to investigate individual sources separately (e.g. volcanic craters, e.g. Tamburello et al., 2013, D'Aleo et al., 2016). The cameras are hence, typically pointed at source-vicinity, where the plumes often show turbulent behaviour, mostly a result of aerodynamic effects and buoyancy. The resulting velocity fields hence, often
deviate significantly from the meteorological background wind field. Luckily, the high resolution of imaging systems allows to account for these fluctuations by directly measuring projected 2D velocity fields, for instance using correlation techniques or optical flow (OF) algorithms (e.g. Krueger et al., 2013, Bjornsson et al., 2013, Peters et al., 2015, Lopez et al., 2015, Stebel et al., 2015).
OF algorithms can detect motion between consecutive frames, by tracking local intensity (or phase) gradients (see Jähne,
1997 for a comprehensive introduction). They typically yield dense displacement vector fields (*DVF*), allowing for gas velocity retrievals down to pixel level. This is especially advantageous under turbulent plume conditions, where correlation based techniques (e.g. McGonigle et al., 2005) often tend to be inapplicable or are accompanying large uncertainties (e.g. Peters et al., 2015). However, OF algorithms intrinsically rely on notable, "trackable" structures (contrast) in the images. As a result, they often tend to fail in low-contrast image areas. To a certain degree, this can be accounted for by increasing the size of
the averaging neighbourhood around each pixel, or by performing a multi-scale analysis (i.e. from coarse to fine features, e.g.





using image pyramids). However, if such "gaps" (of missing structure) become too large, the algorithms will ultimately fail to detect any motion. Kern et al. (2015) also recognise the problem and account for it using reliable motion vectors from other image areas (e.g. plume edges or high contrast plume areas).

We discuss the impact of potential erroneous OF motion vectors on the retrieval of $SO_2$-emission-rates using UV $SO_2$ cameras. We propose a method to correct for this effect, which is based on histograms of the retrieved OF motion fields. The analysis is performed in a localised manner within a specific region-of-interest (ROI) in the images (e.g. in proximity to a plume transect used for the emission-rate analysis). It measures the local predominant gas velocity vector (in the ROI) based on distinct peaks in the histograms. We apply the method to two different volcanic datasets recorded at Mt. Etna, Italy and
Guallatiri, Chile.

    The paper is organised as follows: Sect. 2 starts with a short introduction into the technique of UV $SO_2$ cameras and the required data analysis. Sect. 2.2 provides information about the two datasets (i.e. technical setup, measurement locations), followed by details regarding the image analysis of both datasets (Sect. 2.3). The proposed correction for optical flow based
velocity retrievals is introduced in Sect. 2.4. In Sect. 3 the retrieved $SO_2$-emission-rates for the Etna and Guallatiri datasets are presented and compared to results based on 1. the uncorrected OF *DVF* and 2. assuming a constant plume velocity based on a cross-correlation analysis. A summary and discussion is given in Sect. 4, followed by our conclusions.

## 2   Methodology

### 2.1   UV $SO_2$ cameras

UV $SO_2$ cameras measure plume optical densities (ODs) in two wavelength windows of about $10\,nm$ width using dichroic filters. The two filters are typically centered around 310 nm ($SO_2$ "on-band" filter, i.e. sensitive to $SO_2$ absorption) and, at nearby wavelengths, around 330 nm ($SO_2$ "off-band" filter). The latter is used for a first order correction of aerosole scattering in the plume (e.g. Kern et al., 2010). An apparent absorbance (AA) of $SO_2$ can then be calculated based on the ODs measured in both channels:

$$\tau_{AA} = \tau_{on} - \tau_{off} = \ln\left(\frac{I_0}{I}\right)_{on} - \ln\left(\frac{I_0}{I}\right)_{off}.$$     (1)

    Here, $I$, $I_0$ denotes the measured plume and corresponding background intensities, respectively. Note that all quantities in Eq. 1 are a function of the detector pixel position $i,j$ (e.g. $\tau_{AA} \to \tau_{AA}(i,j)$). The calibration of the measured AA values (i.e. conversion into $SO_2$ column densities $S_{SO_2}(i,j)$) can be performed using $SO_2$ calibration cells or using data from a DOAS spectrometer viewing the plume (Lübcke et al., 2013) or a combination of both. The $SO_2$-emission-rates are typically calculated
along a suitable plume cross section (PCS) $\ell$ in the $SO_2$-CD images $S_{SO2}(i,j)$ (e.g. a straight line) by performing a discrete



integration of the form:

$$\Phi(\boldsymbol{\ell}) = f^{-1} \sum_{m=1}^{M} S_{SO2}(m) \cdot \boldsymbol{v}_{eff}(m) \cdot d_{pl}(m) \cdot \Delta s(m), \qquad (2)$$

where $m$ denotes interpolated image coordinates $(i,j)$ along $\boldsymbol{\ell}$, $f$ is the camera focal length, $d_{pl}$ is distance between the camera and the plume and $\Delta s(m)$ is the integration step length (for details see Gliß et al., 2017). The effective velocity

$$5 \quad \boldsymbol{v}_{eff}(m) = \langle \bar{\boldsymbol{v}}(m) \cdot \hat{\boldsymbol{n}}(m) \rangle \qquad (3)$$

is measured relative to the normal $\hat{\boldsymbol{n}}$ of $\boldsymbol{\ell}$ (i.e. constant in case of straight retrieval lines) using the corresponding velocity vector $\bar{\boldsymbol{v}}(m)$. The velocities, if retrieved from the images, represent averages along the line-of sight (LoS) of each pixel (see e.g. Krueger et al., 2013 for a derivation). Since the velocity components in LoS direction cannot be measured from the images, the measured velocities are approximately underestimated by a factor of $\cos(\alpha)$ ($\alpha$ being the angle between plume and image
10   plane). However, to first order (and at small angles $\alpha$), this cancels out since the length of the LoS inside the plume (and thus, the measured $SO_2$ CDs) increases by approximately the same $\cos(\alpha)$ factor (Mori and Burton, 2006).

## 2.2 Example data

The proposed method to correct for unphysical OF velocity vectors is applied to two volcanic datasets recorded at Mt. Etna (Italy) and Guallatiri volcano (Chile). Both datasets were recorded using a filter-wheel based UV $SO_2$ camera including a
15   DOAS spectrometer. Details about the technical setup for both datasets are summarised in Tab. 1.

**Table 1.** Instrumental setup during both campaigns

|  |  | Etna | Guallatiri |
|---|---|---|---|
| Camera | UV Camera | Hamamatsu C8484-16C | Hamamatsu C8484-16C |
|  | On-band filter | Asahi UUX0310 | Omega Optical, 310BP10 |
|  | Off-band filter | Asahi XBPA330 | Omega Optical, 325BP12 |
|  | UV lens (focal length) | 25 mm | 50 mm |
| DOAS | Spectrometer | Ocean Optics USB 2000+ | Avantes AvaSpec-ULS2048x64 |
|  | T-stabilisation | No (ambient) | $20°$ |
|  | Telescope | $f = 100\,mm, f/4$ | $f = 100\,mm, f/4$ |
|  | Optical fiber | $400\,\mu m$ | $400\,\mu m$ |



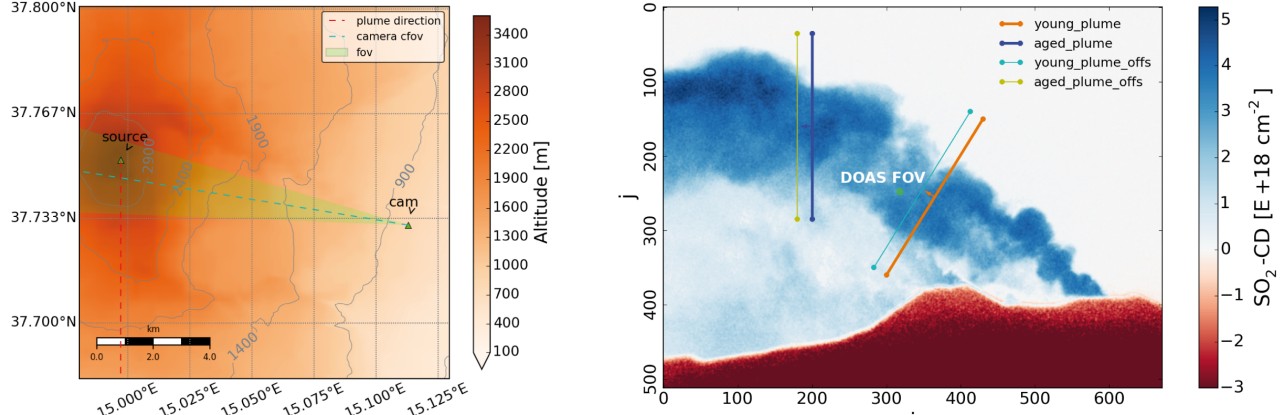

**Figure 1.** (left) Etna overview map showing position and viewing direction of the camera (camera cfov, fov) which was located on a roof-top in the town Milo. Also indicated is the summit area (source) and the plume azimuth (red dotted line). (right) shows an example $SO_2$-CD image of the Etna plume including two PCS lines (orange / blue) used for emission-rate retrievals. Two offset lines (cyan, yellow) are used for cross-correlation based plume velocity retrievals for each PCS line (cf. Appendix B4). Position and extent of the DOAS-FOV for the camera calibration is indicated by a green spot.

### 2.2.1 Etna data

Mt. Etna is a stratovolcano situated in the eastern part of the island of Sicily, Italy. We present a short UV camera dataset recorded on 16.09.2015 between 07:06 – 07:22 UTC (see Tab. 1 for a technical setup of the instruments used for the obser-vations). The data was recorded during a field campaign which took place about 2.5 months prior to a major eruptive event

5   (i.e. in early December 2015, e.g. Smithsonian-Institution, 2013a). The volcano showed quiescent degassing behaviour during all days of the campaign. The measurements were performed from the roof-top of a building located in the town Milo, about 10.3 km from the source. An overview map is shown in Fig. 1.

**Plume conditions**

10   During the 15 minutes of data, the meteorological conditions were stable showing a slightly convective plume of the Etna north-east crater (NEC) advected downwind (into the left image half, cf. Fig. 1). The emissions of the other craters are more diffuse and could not be fully captured since they were partly covered by the volcanic flank. Therefore, we kept the focus on the NEC emissions which were investigated along two example PCS lines located at two different positions downwind the source (orange / blue lines in Fig. 1). A video of the Etna emissions is shown in supplementary video no. 1.





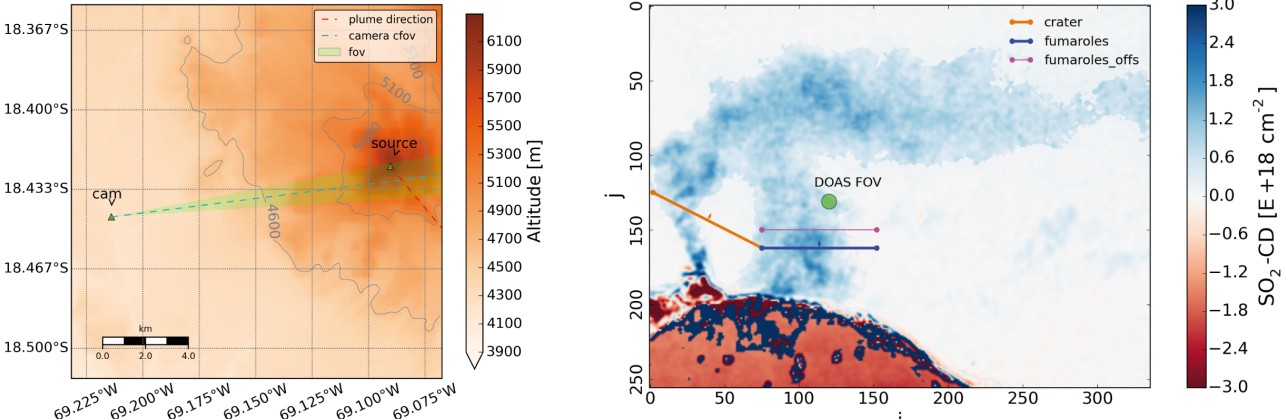

**Figure 2.** (left) Guallatiri overview map showing position and viewing direction of camera (camera cfov, fov), summit area (source) and the plume azimuth (red dotted line). (right) shows an example $SO_2$-CD image of the emissions, including two PCS lines used to retrieve $SO_2$-emission-rates from the central crater (orange) and the fumarolic field (blue). An additional line (magenta) is used to estimate gas velocities using a cross-correlation algorithm (relative to blue PCS line, cf. Appendix B4). Position and extent of the DOAS-FOV is indicated with a green spot in (b).

### 2.2.2  Guallatiri data

Guallatiri ($18°\ 25'\ 00''$ S, $69°\ 5'\ 30''$ W, 6.071 m a.s.l.) is a stratovolcano located in the Altiplano, northern Chile. The last confirmed eruptive events date back to 1960 (Smithsonian-Institution, 2013b). Due to its remote location little is known about the volcano.

5  The presented data is part of a short field campaign between 20 and 22 November 2014. During the three days, the volcano showed quiescent degassing behaviour from the central crater and from a fumarolic field on the SW flank of the volcano. Due to frequent cloud abundances, only a small fraction of the acquired data was suited for the investigation of the $SO_2$ emissions. A cloud free time window between $14{:}48-14{:}59$ UTC on 22/11/2014 was chosen (see Tab. 1 for details about the instrumental setup). An overview map is shown in Fig. 2. The measurements were performed at a distance of $13.3\,\mathrm{km}$ away from the source.

**Plume conditions**

Compared to Etna, the Guallatiri emissions showed rather turbulent behaviour with strong variations in the local velocities. The central crater plume, in particular, changed its overall direction significantly over time which can be seen in supplementary video no. 2. Emission rates were retrieved along two (connected) PCS lines in the young plume shown in Fig. 2. The lines were

15  chosen such that the emissions from the central crater and the fumarolic field could be investigated separately.





## 2.3 Data analysis

The image analysis was performed using the Python software *Pyplis* (Gliß et al., 2017). In a first step, all images were corrected for electronic offset and dark current followed by a first order correction for the signal dilution effect. The latter was applied based on Campion et al., 2015 using suitable volcanic terrain features in the images to retrieve an estimate of the atmospheric scattering extinction coefficients in viewing direction of the camera. The extinction coefficients were used to correct the measured radiances of plume image pixels for the scattering contribution. The latter were identified using an appropriate $\tau$ threshold applied to on-band OD images.

The sky background intensities (required to for the retrieval of AA images, Eq. 1) were determined using on / off sky reference images (SRI) recorded close in time to the plume image data. Variations in the sky background intensities and curvature between the plume images and SRI were corrected both in horizontal and vertical direction using suitable gas (and cloud) free sky reference areas in the plume images. All AA images were corrected for cross detector variations in the $SO_2$ sensitivity using a correction mask calculated from cell calibration data as outlined by Lübcke et al. (2013). The AA images were calibrated using plume $SO_2$-CDs retrieved from a co-located DOAS instrument (cf. Tab. 1, see Sect. B1 for details regarding the DOAS retrieval). Position and extent of the DOAS-FOV within the camera images are shown in Figs. 1 (Etna) and 2 (Guallatiri) and were identified using the Pearson correlation method described in Gliß et al. (2017).

The gas velocities in the plume were retrieved both using the Farnebäck OF algorithm and the cross correlation method outlined in McGonigle et al. (2005). Nonphysical OF motion vectors along the emission-rate retrieval lines were identified and corrected for using the proposed OF histogram method, which is described in Sect. 2.4. Note that for the analysis all images were downscaled by a factor of 2 (using a Gaussian pyramid approach).

### 2.3.1 Etna

The required plume distances for the emission-rate retrieval were derived from the camera location and viewing direction and assuming a meteorological wind direction of $(0 \pm 20)°$ (north-wind, cf. Fig. 1). The latter was estimated based on visual observation. The camera viewing direction was retrieved using the position of the south-east (SE) crater in the images. The signal dilution correction was performed using atmospheric scattering extinction coefficients retrieved 20 minutes prior to the presented observations (i.e. from one on and one off-band image recorded at 06:45 UTC, cf. Fig. 12 in Gliß et al., 2017). During this time the camera was pointed at a lower elevation angle and the images contained more suitable terrain features for the correction. Extinction coefficients of $\epsilon_{on} = 0.0743\,\mathrm{km}^{-1}$ and $\epsilon_{off} = 0.0654\,\mathrm{km}^{-1}$ could be retrieved and were used to correct plume image pixels. The latter were identified from on-band OD images using a threshold of $\tau_{on} = 0.05$. The dilution corrected AA images were calibrated using the DOAS calibration curve shown in Appendix B2. An example calibrated and dilution corrected $SO_2$-CD image is shown in Fig. 1.





### 2.3.2 Guallatiri

The plume distances were retrieved per pixel column assuming a meteorological wind direction of $(320 \pm 15)°$. The latter was estimated based on visual observation combined with a MODIS image (see supplementary material) recorded at 15:05 UTC, in which the plume could be identified. The viewing direction of the camera was retrieved based on the geographical location of the summit area in the images.

The dilution correction was performed using scattering extinction coefficients of $\epsilon_{on} = 0.0855 \pm 0.0012\,\mathrm{km}^{-1}$ and $\epsilon_{off} = 0.0710 \pm 0.0008\,\mathrm{km}^{-1}$. The latter were retrieved between 14:48 – 14:59 UTC using images from a second UV camera, which was equipped with a $f = 25\,\mathrm{mm}$ lens (i.e. a wider FOV) and hence, contained more suitable topographic features for the retrieval. Plume pixels for the dilution correction were identified from on-band OD images using a threshold of $\tau_{on} = 0.02$. An example dilution corrected $SO_2$-CD image is shown in Fig. 2. The DOAS calibration curve is shown in Appendix B2. Fig. 2 shows an example dilution corrected and calibrated $SO_2$-CD image.

### 2.3.3 Radiative transfer effects

Both the Etna and Guallatiri data were recorded at long distances ($> 10$km). Consequently, the applied dilution correction accompanies relatively large uncertainties of statistical nature, which we estimate to $\pm 50\%$, based on Campion et al. (2015). Furthermore, in-plume radiative transfer (e.g. multiple scattering due to aerosols, $SO_2$ saturation, see e.g. Kern et al., 2013) may have affected the results to a certain degree. However, both plumes showed only little to no condensation. We therefore assess the impact of aerosole multiple scattering negligible. In the case of Etna, $SO_2$ saturation around 310 nm may induce a small systematic underestimation in the $SO_2$-emission-rates. This is due to the comparatively large observed $SO_2$-CDs of up to $5 \cdot 10^{18}\,\mathrm{cm}^{-2}$. The impact of $SO_2$ saturation is, however likely compensated to a certain degree, since the DOAS $SO_2$-CDs (used to calibrate the camera) were retrieved at less affected wavelengths between $\Delta\lambda_0 \approx (315 - 326)\,\mathrm{nm}$ (cf. Appendix B1). The same fit-interval is used in Gliß et al. (2015) who performed MAX-DOAS measurements of the Etna plume under comparable conditions. They account for $SO_2$ saturation by using the weak $SO_2$-bands between $\Delta\lambda_1 \approx (350 - 373)\,\mathrm{nm}$ (see also Bobrowski et al., 2010) and find relative deviations of about 10 % between the two wavelength ranges and for $SO_2$-CDs exceeding $5 \cdot 10^{18}\,\mathrm{cm}^{-2}$ (i.e. $\frac{\Delta\lambda_0}{\Delta\lambda_1} \approx 0.9$ cf. Fig. A3 in Gliß et al., 2015). We therefore estimate the impact of $SO_2$ saturation to be below 20 % for our data.

### 2.4 Optical flow histogram analysis

We developed a method to improve optical flow (OF) based gas velocity retrievals needed for the analysis of $SO_2$-emission-rates (Eq. 2) using UV camera systems. The OF analysis of an image pair yields dense displacement vector fields (*DVF*'s) of the observed gas plumes. In some areas of the image, the *DVF* represents the actual physical motion of gas in the plume, while other image areas may contain unphysical motion vectors (e.g. in low-contrast plume regions). The proposed method aims to identify all successfully constraint motion vectors and from these, derives an estimate of the average (or predominant) velocity





vector in the plume. The latter is then used to replace unphysical motion vectors in the *DVF*. We recommend to perform the analysis in a localised manner, within a specific region-of-interest (ROI) since the velocity fields can show large fluctuations over the entire image (e.g. change in direction or magnitude).

Fig. 3 shows an example *DVF* (left) retrieved from the Etna plume including an example rectangular ROI (top). Two further

images show the corresponding OF displacement orientation angles $\varphi$ (middle) and flow vector magnitudes $|\boldsymbol{f}|$ (bottom). Histograms $\mathcal{M}$ (i.e. $\mathcal{M}_\varphi$, $\mathcal{M}_{|\boldsymbol{f}|}$) of the motion field are plotted in the right panels, respectively, and were calculated considering all image-pixels belonging to the displayed ROI. From the images and histograms, certain characteristics become clear:

1. Image regions containing unphysical motion estimates are characterised by (local) random orientation and short flow vectors (cf. sky background pixels).

2. These unphysical motion vectors manifest as constant offset in $\mathcal{M}_\varphi$ and as peak at the lower end of $\mathcal{M}_{|\boldsymbol{f}|}$.

3. Image regions showing reliable motion estimates, on the other hand, are characterised by (locally) homogeneous orientation $\varphi$ and magnitudes $|\boldsymbol{f}|$ exceeding a certain minimum length $|\boldsymbol{f}|_{\min}$.

4. These successfully constraint motion vectors manifest as distinct peaks in $\mathcal{M}_\varphi$ and $\mathcal{M}_{|\boldsymbol{f}|}$.

5. The width of these peaks can be considered a measure of the local fluctuations, or the variance, of the velocities (e.g. a

very narrow and distinct peak in $\mathcal{M}_\varphi$ would indicate a highly directional movement).

Based on these histogram peaks, the proposed method derives the local *predominant displacement vector (PDV)* $\overline{|\boldsymbol{f}|}$ . A detailed mathematical description of the analysis is provided in Appendix A. In the following, the most important steps of the analysis are described.

The retrieval of the *PDV* starts with a peak analysis of $\mathcal{M}_\varphi$ and investigates whether a distinct and unambiguous peak can

be identified in the histogram. If this is the case, the expectation value for the local movement direction $\varphi_\mu$ and the angular confidence interval $\mathcal{I}_\varphi$ are retrieved based on the position and the width of the main peak in $\mathcal{M}_\varphi$ (using the 1st and 2nd moments of the distribution). The analysis of $\mathcal{M}_\varphi$ involves a peak-detection routine based on a Multi-Gauss parametrisation. The latter is done to ensure that the retrieved parameters $\varphi_\mu$ and $\mathcal{I}_\varphi$ are not falsified due to potential additional peaks in the distribution (e.g. a cloud passing the scene, e.g. illustrated in Fig. 12). Based on the analysis of $\mathcal{M}_\varphi$, a second histogram

$\mathcal{M}_{|\boldsymbol{f}|}$ is determined, containing the displacement magnitudes $|\boldsymbol{f}|$ of all vectors matching the angular confidence interval $\mathcal{I}_\varphi$ and exceeding the required minimum magnitude $|\boldsymbol{f}|_{\min}$. Also here, an expectation value $|\boldsymbol{f}|_\mu$ and confidence interval $\mathcal{I}_{|\boldsymbol{f}|}$ are estimated based on the 1st and 2nd moments of the histogram.

The analysis yields four parameters $p_{\text{ROI}} = (\varphi_\mu, \varphi_\sigma, |\boldsymbol{f}|_\mu, |\boldsymbol{f}|_\sigma)$ which are used to calculate the predominant displacement vector (*PDV*) within the corresponding ROI:

$$\bar{\boldsymbol{f}}(\text{ROI}) = [|\boldsymbol{f}|_\mu \cdot \sin \varphi_\mu, |\boldsymbol{f}|_\mu \cdot \cos \varphi_\mu]^T. \qquad (4)$$





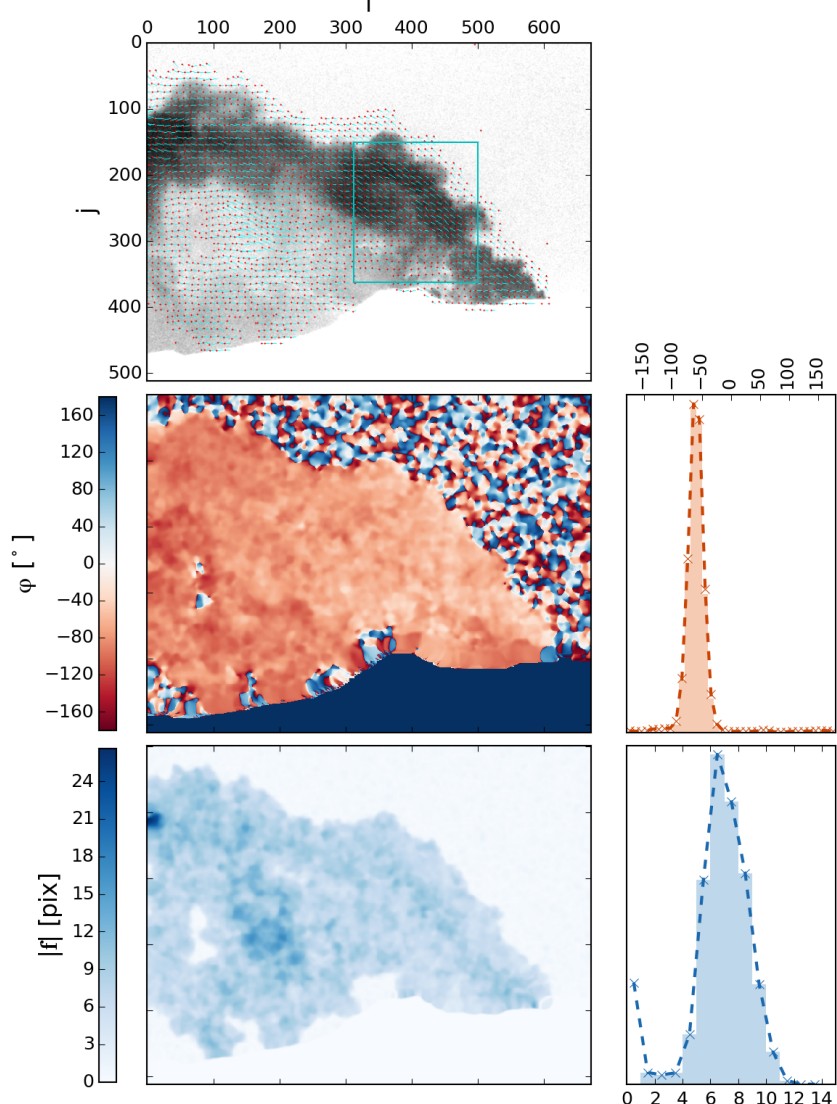

**Figure 3.** Example output of the Farnebäck optical flow algorithm including a rectangular ROI (top). Two further images show the corresponding orientation angles $\varphi$ (middle) and magnitudes $|\boldsymbol{f}|$ (bottom) of the *DVF*. Corresponding histograms $\mathcal{M}_\varphi$ and $\mathcal{M}_{|\boldsymbol{f}|}$ are plotted on the right, respectively, and include all pixels in the displayed ROI. The histograms are plotted both including (dashed lines) and excluding (red / blue shaded areas) short flow vectors (i.e. $|\boldsymbol{f}| > |\boldsymbol{f}|_{\min} = 1.5$pix. The orientation angles are plotted in an interval $-180° \leq \varphi \leq +180°$ where $-90°$, $+90°$ correspond to $-i$, $+i$ directions, respectively and $0°$ to the vertical upwards direction ($-j$). The *DVF* was calculated using two consecutive AA images ($\Delta t = 4.0s$) of the Etna plume, recorded on 16/09/2015 at 07:14 AM.

The projected plume velocity vector for the ROI can then be calculated as:

$$\bar{\boldsymbol{v}}(\mathrm{ROI}) = \bar{\boldsymbol{f}}(\mathrm{ROI}) \cdot \frac{d_{\mathrm{pl}}\Delta_{\mathrm{pix}}}{f \cdot \Delta t}, \tag{5}$$





where $f$ and $\Delta_{\text{pix}}$ denote lens focal length and the pixel pitch of the detector, and $d_{\text{pl}}$ is the distance between the camera and the plume. Ill-constraint motion vectors in the *DVF* can then be identified with a certain confidence based on $\mathcal{M}_\varphi$ and $\mathcal{M}_{|\boldsymbol{f}|}$. In this article, the method is demonstrated using the OpenCV (Bradski, 2000) Python implementation of the Farnebäck OF algorithm (Farnebäck, 2003, also used in Peters et al., 2015). It is pointed out, though, that it can be applied to *DVFs* from any
motion estimation algorithm.

### 2.4.1   Applicability and uncertainties

The proposed method offers an efficient solution to identify flow vectors containing actual gas movement and separate them from unphysical results in the *DVF*. The method is based on a local statistical analysis of the histograms $\mathcal{M}_\varphi$ and $\mathcal{M}_{|\boldsymbol{f}|}$. A number of quality criteria were defined in order to ensure a reliable retrieval of the local displacement parameters:

C1: A minimum fraction $r_{\text{min}}$ of all pixels in the considered ROI is required to exceed the minimum magnitude $|\boldsymbol{f}|_{\text{min}}$. The latter can, for instance, be set equal one or can be estimated based on the flow vector magnitudes retrieved in a homogeneous image area (e.g. randomly oriented sky background areas in Fig. 3).

C2: The same minimum fraction $r_{\text{min}}$ of pixels is required to match the angular expectation range specified by $\mathcal{I}_\varphi$ (at a certain confidence level $n\sigma$, cf. Appendices A and B3).

C3: If additional peaks are detected in $\mathcal{M}_\varphi$, they are required to stay below a certain significance value $\mathcal{S}$. The latter is measured relative to the main peak based on the integral values (cf. Appendix A3 and Fig. 12).

If any of these constraints cannot be met, the analysis is aborted. The settings used in this study are summarised in Tab. 2.

Please note that the method cannot account for any uncertainties intrinsic to the used OF algorithm since these directly
propagate to the derived histogram parameters. It is therefore recommended, to assess the general performance of the used OF algorithm independently and before applying the histogram correction. For the latter, optical flow inter-comparison benchmarks (e.g. Baker et al., 2011, Menze and Geiger, 2015) can provide useful information to assess the performance (e.g. accuracy) and applicability (e.g. computational demands) of individual OF algorithms. The Farnebäck algorithm used in this study showed sufficient performance both in Peters et al. (2015) and in the KITTI benchmark (cf. Menze and Geiger, 2015). The latter find
that the algorithm yields correct velocity estimates in about 50 % of all cases. Here, "correct" means, that the disparity between a retrieved flow-vector endpoint and its true value does not exceed a threshold of 5 %. Based on these findings, we assume a conservative intrinsic uncertainty of 15% for the effective velocities (Eq. 3) retrieved from reliable OF motion vectors. For the unphysical motion vectors (which are replaced by the *PDV*) we assume a conservative uncertainty based on the width of the histogram peaks.
Finally, we point out that the proposed histogram correction does not constitute any significant additional computational demands. For our data (i.e. $1344 \times 1024\,\text{pix}$) and on an Intel i7, 2.9 GHz machine, the required computation time for the correction





is typically less than $0.1\,\mathrm{s}$. In contrast, the Farnebäck OF algorithm itself typically requires $1.5\,\mathrm{s}$ (same specs.) and can be considered fast, in comparison with other solutions (e.g. Baker et al. (2011)).

## 3 Results

The new method was applied to the Etna and Guallatiri datasets introduced in Sect. 2.2. $SO_2$-emission-rates (Eq. 2) of both
sources were retrieved as described in Sect. 2.3 along the corresponding PCS lines (cf. Figs. 1 and 2). In order to assess the
performance of the proposed correction we use the following three methods to estimate the gas velocities in the plume:

1. *glob*: based on cross-correlation analysis at position of PCS line $\ell$ (i.e. the estimated velocity is applied to all pixels on $\ell$ and to all images of the time series).

2. *flow_raw*: using raw output from the Farnebäck algorithm (i.e. without correction for erroneous flow vectors).

3. *flow_hybrid*: using reliable optical flow vectors, identify and replace unphysical vectors using the *DVF* from the histogram analysis.

Tab. 2 (in Appendix B3) summarises all relevant settings for the OF based velocity retrievals. Note, that the required minimum magnitude for successfully constrained motion vectors was set per image and ROI using the lower end of $\mathcal{I}_{|\boldsymbol{f}|}$ at $1\sigma$ confidence. In order to assess the impact of unphysical motion vectors on the retrieved $SO_2$-emission-rates, we define the ratio $\kappa$:

$$\kappa = \frac{\chi_{\mathrm{pix\,ok}}}{\chi_{\mathrm{all}}}, \tag{6}$$

where $\chi_{\mathrm{all}}$ corresponds to the $SO_2$ ICA considering all pixels on $\ell$ and for $\chi_{\mathrm{pix\,ok}}$ only pixels showing reliable flow vectors
(on $\ell$) are considered. $\kappa = 1$, for instance, means that all motion vectors on $\ell$ are considered reliable. The ROIs for the OF
histogram analysis were defined for each PCS line individually (based on the position and orientation of the line).

### 3.1 Etna

The OF gas velocities in the plume were calculated from on-band OD ($\tau_{\mathrm{on}}$) images. Fig. 4 shows an example *DVF* of the Etna
plume and the corresponding histograms $\mathcal{M}_\varphi$ and $\mathcal{M}_{|\boldsymbol{f}|}$. Along the orange line, the OF algorithm performs considerably well
with only 7 % of the velocity vectors found ill-constraint. If not corrected for, these unphysical motion vectors would result
in an underestimation of only 1 % in the $SO_2$-emission-rates. For the blue line, on the contrary, a total of 45 % of the pixels
on $\ell$ were found unreliable. Moreover, many of these are located in regions showing large $SO_2$-CDs. Hence, the impact is
considerably large and, if not corrected for, would induce an underestimation of 33 % in the $SO_2$-emission-rates .




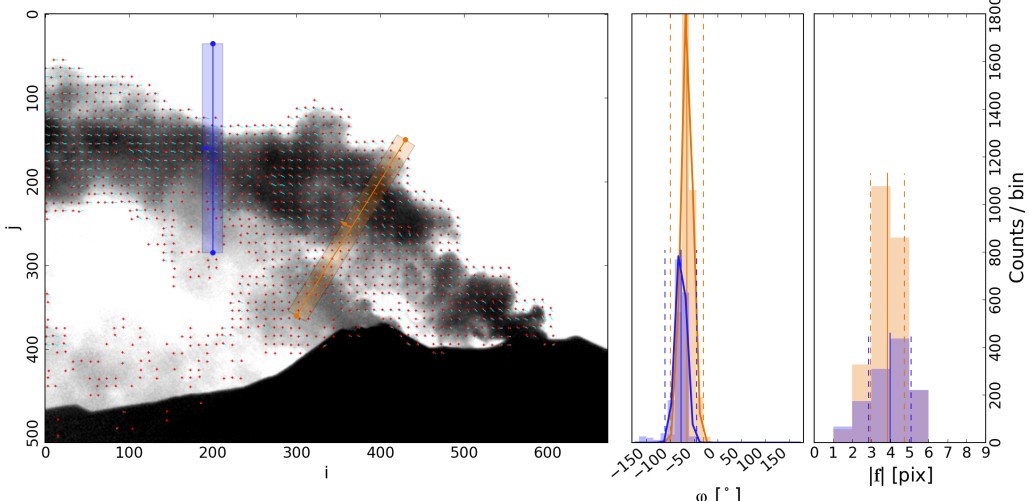

**Figure 4.** (left) Example output of the Farnebäck optical flow algorithm for the Etna plume at 07:13 UTC including the two PCS lines (blue / orange) and the corresponding ROIs used for the histogram analysis (semi-transparent rectangles). (middle) orientation histograms $\mathcal{M}_\varphi$ for both lines (bar plot) including fit results of the Multi-Gauss peak detection (thick solid lines) and (right) magnitude histograms $\mathcal{M}_{|\boldsymbol{f}|}$ determined using condition S7 in Appendix A1. The retrieved histogram parameters $(\varphi_\mu, |\boldsymbol{f}|_\mu)$ and expectation intervals $\mathcal{M}_\varphi$, $\mathcal{M}_{|\boldsymbol{f}|}$ are indicated with solid and dashed vertical lines, respectively. From the corrected *DVF*, average effective velocities of $\boldsymbol{v}_{\text{eff}} = (3.9 \pm 0.5)\,\text{m/s}$ (orange line) and $\boldsymbol{v}_{\text{eff}} = (4.4 \pm 0.8)\,\text{m/s}$ (blue line) were retrieved. Note that *DVF* vectors shorter than one pixel are not plotted.

Prior to the emission-rate analysis, the proposed histogram method was applied to all $\tau_{\text{on}}$ images in order to retrieve time series of the four correction parameters $p = (\varphi_\mu, \varphi_\sigma, |\boldsymbol{f}|_\mu, |\boldsymbol{f}|_\sigma)$. Missing data points (i.e. where the required constraint parameters were not met, cf. Sect. 2.4.1) were interpolated. The results were averaged in time using a combined median filter of width 3 (to remove outliers) and a Gaussian filter ($\sigma = 5$, to remove high frequency variations in the retrieved *DVF*'s).

The results of this pre-analysis are shown in Fig. 5. Due to the stable meteorological conditions the retrieved parameters show only little variation with average values of $\overline{\varphi_\mu} = (-58.3 \pm 5.1)°$ and $\overline{|\boldsymbol{f}|_\mu} = (0.93 \pm 0.09)\,\text{pix/s}$ (orange line) and $\overline{\varphi_\mu} = (-78.5 \pm 3.1)°$ and $\overline{|\boldsymbol{f}|_\mu} = (1.04 \pm 0.06)\,\text{pix/s}$ (blue line).

Fig. 6 shows the results of the $SO_2$-emission-rate analysis for both PCS lines and the three different velocity retrieval

methods. Further included are the corresponding effective velocities (average along $\ell$, 2nd panel) and the retrieved $\kappa$ values (Eq. 6). The latter indicates the percentage impact of unphysical OF motion vectors on the $SO_2$-emission-rates. The plotted uncertainties in the $SO_2$-emission-rates and the effective velocities (shaded areas) were calculated as described in Appendix B5.

$SO_2$-emission-rates between $4.9 - 9.7\,\text{kg/s}$ (average: $7.1\,\text{kg/s}$) and $4.8 - 10.7\,\text{kg/s}$ (average: $7.8\,\text{kg/s}$) were retrieved along

the orange and blue line, respectively, using the proposed *flow_hybrid* method. The slightly higher values in the aged plume are





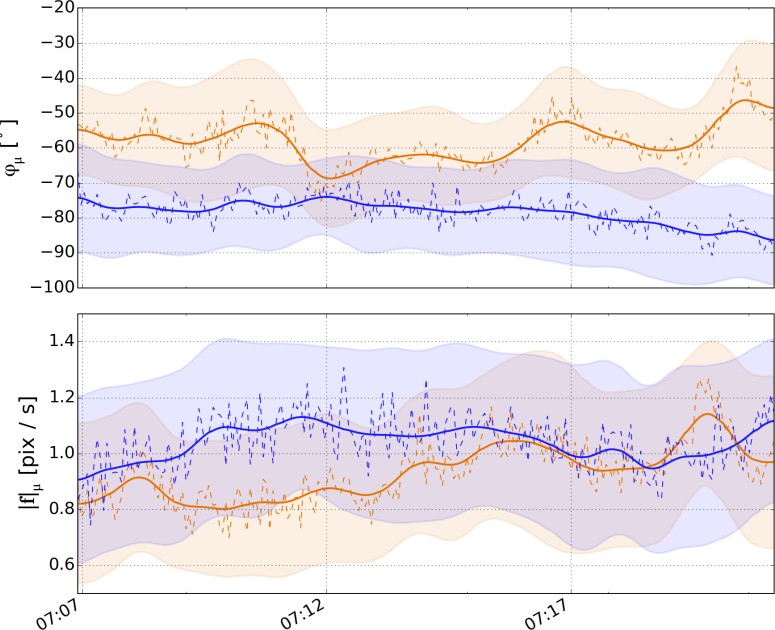

**Figure 5.** Time series of retrieved *PDV* parameters $\varphi_\mu$ and $|\boldsymbol{f}|_\mu$ (dashed lines) for the two Etna PCS lines (same colours, cf. Fig. 1) and corresponding values after applying interpolation and smoothing (solid lines). The expectation intervals $\mathcal{I}_\varphi$ and $\mathcal{I}_{|\boldsymbol{f}|}$ are plotted as shaded areas.

likely due to the fact, that this line captures more of the emissions from the other Etna craters (cf. supplementary video no. 1). The corrected OF emission-rates show good agreement with the results using the cross-correlation velocities (*glob* method). The latter, however, tend to be slightly increased by about $+14\%$ (cf. Fig. 7). The *flow_raw* method (i.e. uncorrected OF velocities), on the contrary, often yields significantly decreased $SO_2$-emission-rates, especially in situations where unphysical OF

5   motion vectors coincide with either of the retrieval lines (i.e. low $\kappa$ value, cf. Fig. 4). The latter show rather strong fluctuations between consecutive frames (i.e. local scatter in the $\kappa$ values) with an average impact of $\overline{\kappa} = (0.68 \pm 0.15)$. These fluctuations are due to the somewhat random nature of the initial problem. Namely, that the occurrence (and position) of regions containing unphysical motion vectors can change significantly between consecutive frames (cf. Fig. 4). These *unphysical* fluctuations (in the estimated gas velocities) directly propagate to the $SO_2$-emission-rates (retrieved using the *flow_raw* method) and may thus,

10   not to be misinterpreted with actual (high-frequency) variations in the $SO_2$-emission-rates.

Relative deviations of the three methods are shown in Fig. 7 (normalised to the results from the proposed *flow_hybrid* method). The cross-correlation based retrievals (*glob*) tend to yield slightly larger $SO_2$-emission-rates (by $+14\%$ on average) while the uncorrected OF (*flow_raw*) often shows underestimated results (by $-20\%$ on average). However, we point out again, that the these underestimations generally show a rather strong variability. This includes cases showing considerably large un-





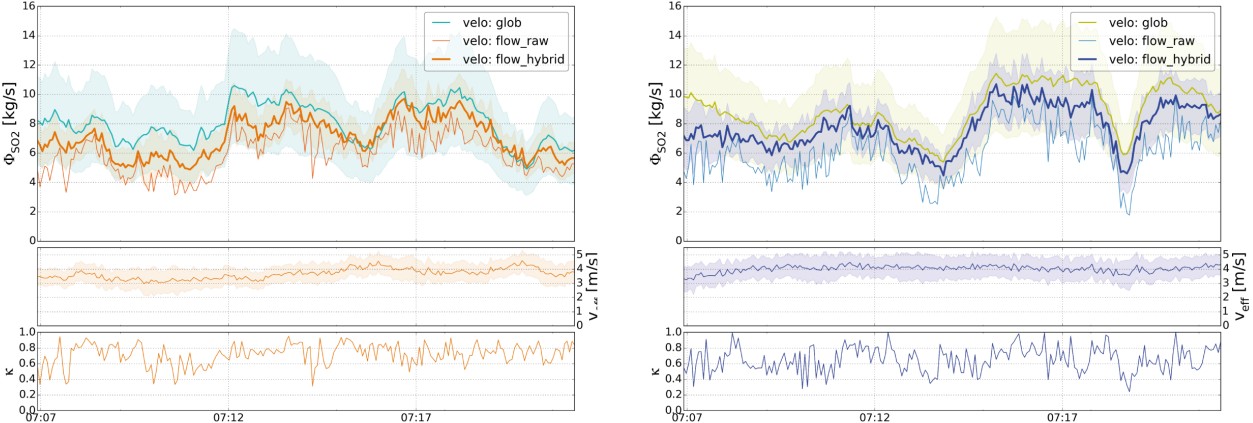

**Figure 6.** Time series of Etna emission-rates (top panel), showing emissions of the young (left) and the aged (right) plume of the NE crater (orange / blue colours, respectively) using the two PCS lines shown in Fig. 2. Emission rates were retrieved using the three different velocity retrieval methods described above. Uncertainties (shaded areas) are only plotted for the *flow_hybrid* method and the cross-correlation method ("*glob*"). Further included are time series of effective velocities (Eq. 3, middle panel) and $\kappa$ values (Eq. 6, bottom) retrieved from the proposed histogram analysis.

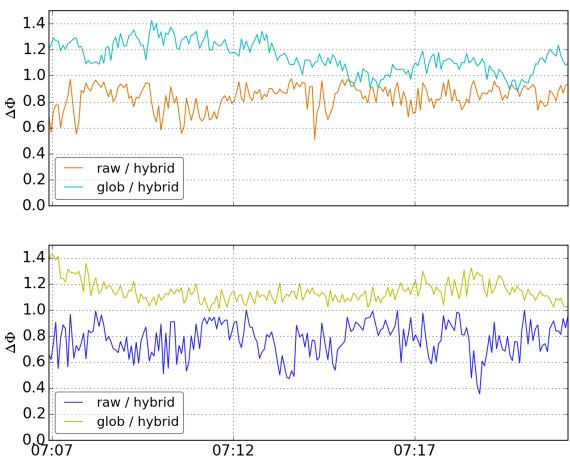

**Figure 7.** Relative deviations of retrieved $SO_2$-emission-rates shown in Fig. 6 for the "young_plume" (top) and the "aged_plume" (bottom) PCS lines using the same colour codes as in Fig. 6. The ratios are plotted relative to the results of the proposed *flow_hybrid* method. Results based on the cross-correlation analysis tend to be slightly larger (by about $+14\%$) while the uncorrected OF velocities often yield underestimated $SO_2$-emission-rates (up to 62%).



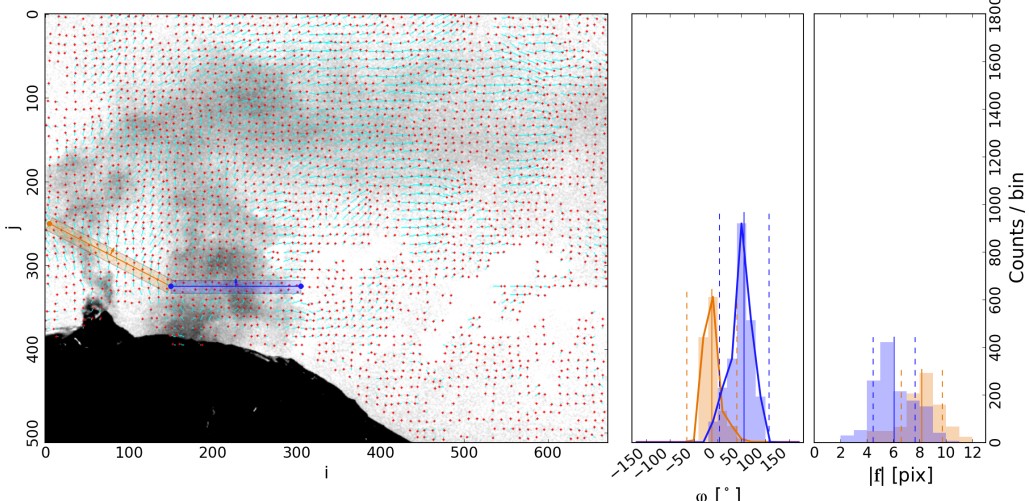

**Figure 8.** Example output of the Farnebäck optical flow algorithm for Guallatiri (left) including the two example PCS lines (blue / orange line) and ROIs. The histograms $\mathcal{M}_\varphi$ (middle) and $\mathcal{M}_{|\boldsymbol{f}|}$ (right) were used to retrieve the expectation intervals $\mathcal{I}_\varphi$ and $\mathcal{I}_{|\boldsymbol{f}|}$ and the corresponding *PDV* (cf. Eq. 4). From the latter, effective velocities of $\boldsymbol{v}_{\mathrm{eff}} = (3.1 \pm 0.5)\,\mathrm{m/s}$ (crater) and $\boldsymbol{v}_{\mathrm{eff}} = (1.8 \pm 0.6)\,\mathrm{m/s}$ (fumaroles) were retrieved. The two images used to calculate the displayed *DVF* were recorded at 14:49 UTC. Note that motion-vectors shorter than 1 pixel are not plotted.

derestimations (up to 62%) and other cases, where the OF algorithm appears to perform sufficiently (i.e. $\Delta\Phi = 1$ in Fig. 7).

## 3.2 Guallatiri results

The OF gas velocities for the Guallatiri data were retrieved using the on-band OD images. An example *DVF* is shown in Fig. 8.

5   Here, the two sources are clearly separable, showing a convective central crater plume (approx. location at cols. $i \approx 50 - 80$) and the emissions from the fumerolic field located behind the volcanic flank ($i \approx 100 - 300$). Further included are the results of the proposed OF histogram analysis, which was performed relative to the two displayed PCS lines used for the SO$_2$-emission-rate analysis (cf. Fig. 2).

In this example, the OF algorithm performed considerably well. The uncorrected OF would therefore result in a small under-

10  estimation of 6 % (crater) and 3 % (fumaroles) in the SO$_2$-emission-rates. The different plume characteristics of both sources can be clearly identified based on the displayed histogram distributions. The central crater plume (orange colors) is almost vertically rising ($\varphi = (-12.4 \pm 17.5)°$) and reaches velocity magnitudes of up $|\boldsymbol{v}|_{\mathrm{max}} = 4.5\,\mathrm{m/s}$. The fumarolic emissions (blue colors) are less convective ($\varphi = (+55.6 \pm 17.4)°$ ) and show slightly smaller velocities with maximum magnitudes of $|\boldsymbol{v}|_{\mathrm{max}} = 3.4\,\mathrm{m/s}$.





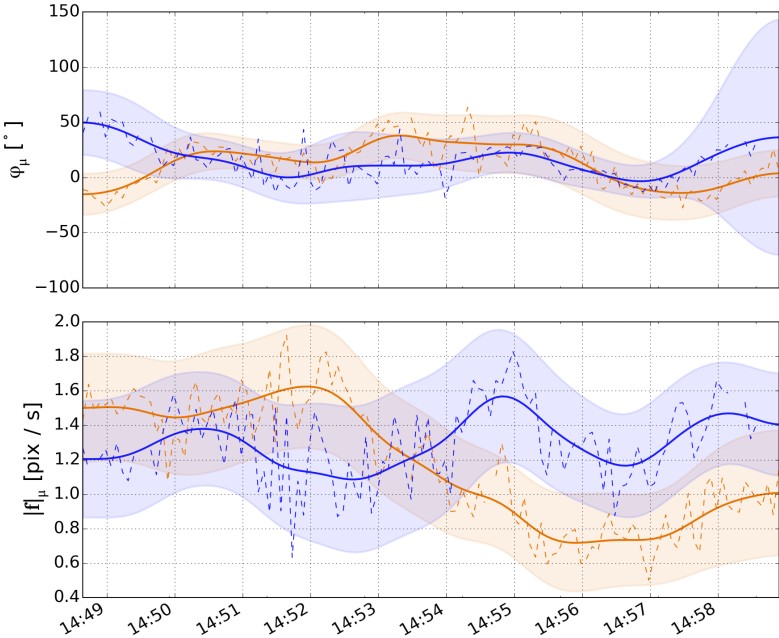

**Figure 9.** Time series of retrieved *PDV* parameters $\varphi_\mu$ and $|\boldsymbol{f}|_\mu$ (dashed lines) for the two Guallatiri PCS lines (same colours, cf. Fig. 8) and the corresponding values after applying interpolation and smoothing (solid lines). The expectation intervals $\mathcal{I}_\varphi$ and $\mathcal{I}_{|\boldsymbol{f}|}$ are plotted as shaded areas.

The time series of the interpolated and smoothed displacement parameters for both PCS lines is shown in Fig. 9. Compared to Etna, the two plumes show considerably more variability both in orientation and in the velocity magnitudes (cf. Figs. 5 and 9). The resulting average values are $\overline{\varphi_\mu} = (12.6 \pm 16.8)°$ and $\overline{|\boldsymbol{f}|_\mu} = (1.17 \pm 0.33)\,\text{pix}/\text{s}$ (crater) and $\overline{\varphi_\mu} = (15.9 \pm 13.1)°$ and $\overline{|\boldsymbol{f}|_\mu} = (1.30 \pm 0.13)\,\text{pix}/\text{s}$ (fumaroles). Due to the rather strong temporal variations, the emissions of both sources could

not always be successfully separated using the two (fixed) PCS lines. This can be seen in supplementary video no. 2, which shows the evolution of $SO_2$-emission-rates for both PCS lines.

The results of the emission-rate analysis are shown in Fig. 10, again, including effective velocities and $\kappa$ values for both PCS lines (cf. Fig. 6). As in the Etna example, the $SO_2$-emission-rates were calculated using the three different velocity retrieval methods introduced above (i.e. *glob*, *flow_raw*, *flow_hybrid*). In general, similar trends can be observed. The uncorrected

OF often causes significant underestimations in the $SO_2$-emission-rates. It furthermore accompanies rather strong (and unphysical) high-frequency fluctuations which are propagated to the $SO_2$-emission-rates (see Sect. 3.1 for a discussion). The cross-correlation based results generally show good agreement with the *flow_histo* method. However, they have to be interpreted carefully, due to the highly turbulent plume conditions. Especially the central crater plume showed large variations both in the overall gas movement direction $\varphi$ and in the velocity magnitudes (cf. Fig. 5 and suppl. video no. 2). As a result, the

cross-correlation analysis could only be successfully applied to the fumarole emissions (cf. Appendix B4). Therefore, the same





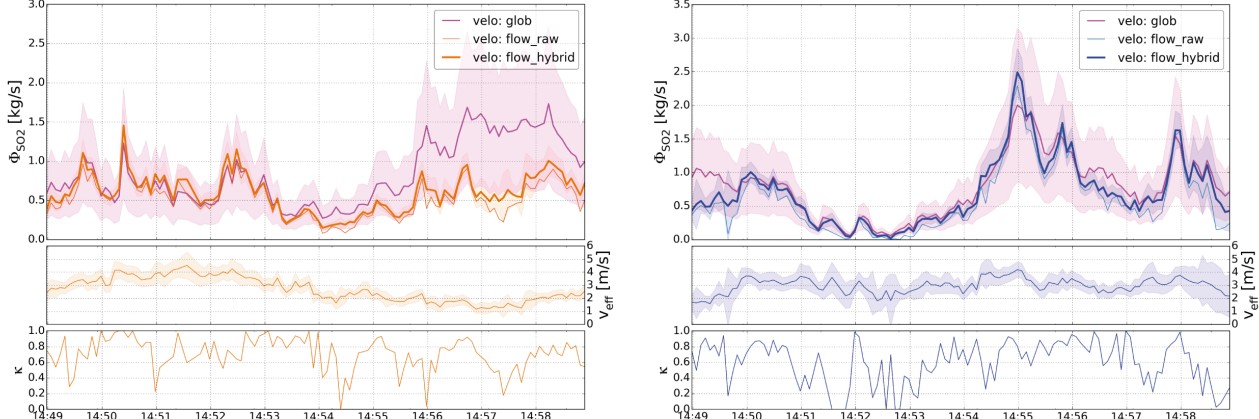

**Figure 10.** Guallatiri SO$_2$-emission-rates from the summit crater (left, orange colours) and the fumarolic field (right, blue colours) using the two PCS lines shown in Fig. 2. Uncertainties (shaded areas) are only plotted for the *flow_hybrid* and the *glob* (cross-correlation based velocity) retrieval methods. Further included are the corresponding effective velocities (from the *flow_hybrid* method) and the OF quality factors $\kappa$ (Eq. 6). The central crater emissions show only little variability ($\Phi \approx 0.6\,\mathrm{kg/s}$). The deviations to the cross-correlation method (at the end of the time series) are due to the latter (for details see text). The fumarolic emissions are characterised by a comparatively strong emission "event" at 14:55 UTC showing peak emissions of $2.5\,\mathrm{kg/s}$.

velocity was assumed for the (highly variable) central crater plume. This simplified assumption (of a globally constant gas velocity) resulted in considerably large deviations between the observed SO$_2$-emission-rates (cf. *glob* and *flow_hybrid* in Fig. 10, left, i.e. after 14:54 UTC). We attribute these deviations to the gradual decrease in the effective gas velocities, which could be observed with the OF algorithm (cf. Fig. 10, left, second panel) but was disregarded by the cross-correlation analysis.

The SO$_2$-emission-rates, which were calculated based on the proposed *flow_hybrid* method, show only little variation in the central crater emissions with values ranging between $0.1 - 1.5\,\mathrm{kg/s}$ (Fig. 10, left). The corresponding fumarole emissions, however, show rather strong variations with peak emission-rates of $2.5\,\mathrm{kg/s}$ (at 14:55 UTC), even exceeding the observed central crater amounts. The sum of both sources yields total SO$_2$-emission-rates of $\overline{\Phi}_{\mathrm{tot}} = 1.3 \pm 0.5\,\mathrm{kg/s}$ with peak emissions

10 up to $2.9\,\mathrm{kg/s}$.

Relative deviations of the retrieved SO$_2$-emission-rates between the three velocity methods are shown in Fig. 11. As in the case of Etna, the cross-correlation based results (*glob*) tend to be slightly increased (here: $+35\,\%$) while the uncorrected OF (*flow_raw*) results in an average underestimation of $-20\,\%$.



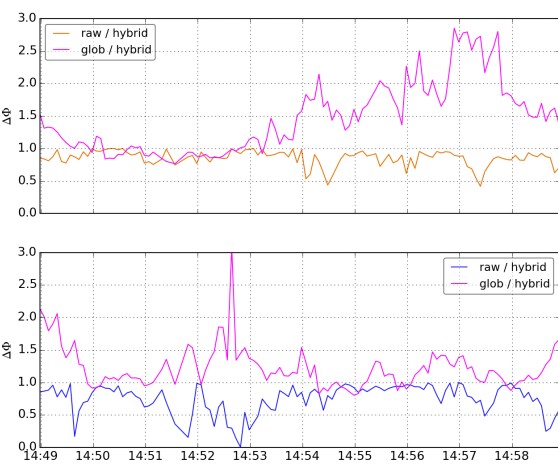

**Figure 11.** Relative deviations of Guallatiri emission-rates shown in Fig. 6. The deviations are plotted as ratios normalised to the results from the proposed method (*flow_hybrid*), both for the crater (top) and the fumarolic (bottom) emissions. Average ratios for cross-correlation based retrievals are $1.48 \pm 0.54$ (crater) and $1.23 \pm 0.32\%$ (fumaroles) and for the *flow_raw* method $0.85 \pm 0.12$ (crater) and $0.75 \pm 0.22\%$ (fumaroles). Again, the latter show a rather strong variability between the images.

## 4 Summary and discussion

### 4.1 Corrected gas velocities

The proposed histogram correction could be successfully applied to the two example datasets from Mt. Etna and Guallatiri. Especially the rather turbulent Guallatiri case clearly demonstrated the necessity of *localised* gas velocity retrievals (both in the spatial and temporal domain). We showed, that OF algorithms are (generally) able to resolve the 2D velocity fields in great detail. However, we also showed that unphysical OF motion vectors often induce significant underestimations in the retrieved $SO_2$-emission-rates. For both datasets, the proposed histogram method was able to account for this issue and resulted in more robust $SO_2$-emission-rate retrievals (cf. Figs. 7 and 11). The corrected results show good coincidence with $SO_2$-emission-rates based on the assumption of a global constant velocity (retrieved using a cross-correlation algorithm). However, the limitations of the cross-correlation method could be clearly demonstrated in the case of the turbulent Guallatiri plume.

### 4.2 Retrieved $SO_2$-emission-rates

The retrieved Etna emission rates of $\sim 8\,\mathrm{kg/s}$ ($\sim 700\,\mathrm{t/d}$) are at the lower end of typically observed values ($> 1000\,\mathrm{t/d}$, e.g. Salerno et al., 2009), ranging from a few up to several hundred $\mathrm{kg/s}$, dependent on the activity (e.g. Edner et al., 1994, D'Aleo et al., 2016). The comparatively low values may be partly due to the fact, that mainly the emissions of the NEC were captured, which nonetheless appeared to be the strongest source during the observation. The presented time series of about $15\,\mathrm{min}$





duration is too short to infer any reliable conclusions related to the state of activity. Nonetheless, it may be noted, that the measurements were recorded about two months prior to a major eruptive event, and that indications of decreased pre-eruptive $SO_2$ emissions have been observed before at Mt. Etna (e.g. Caltabiano et al., 1994). A longer record of Etnas $SO_2$ emissions (i.e. during the months of Sept.-Dec. 2015) would hence be desirable, in order to evaluate whether these low emission-rates

were characteristic for the time period prior to the eruption.

In the case of Guallatiri, no reports are available yet related to its $SO_2$ emissions. This makes the retrieved emission rates of $\sim 1.5\,\mathrm{kg/s}$ (peak: $\sim 3.0\,\mathrm{kg/s}$) an important finding of this study. However, also in this case, the presented time series is rather short and hence not suited to infer typical emission characteristics of the volcano. Future investigations are highly desirable

in order to infer more detailed information related to the emission characteristics of the volcano (e.g. long term averages). Conducting these is obviously more challenging than in the Etna case due to the remote location of the volcano. Space based observations of this considerably weak source may be an option in the future, but appear to be difficult with currently available instrumentation (e.g. Carn et al., 2016).

**5   Conclusions**

In this article, a new method for image based gas velocity measurements in plumes was presented. The success of the method lies in the extraction of quantitative information about gas dynamics inside an emission plume by using the physical information present in a remotely recorded video-image-sequence. The method is based on local gas velocity retrievals using optical flow (OF) algorithms. OF algorithms are a powerful tool for measuring the plume velocities in great detail. However, an intrin-

sic weakness of such algorithms is that they rely on contrast in the images. Hence, they often yield unphysical motion estimates in low-contrast image regions (e.g. in the center of an extended, homogeneous plume). We showed, that this weakness is unacceptable for applications relying on accurate velocity measures at specific image coordinates (such as the discussed application of camera based $SO_2$-emission-rate retrievals).

The proposed method aims to address this issue based on a local post-analysis of an OF displacement-vector-field (*DVF*).

The central idea is to separate reliable from unreliable motion vectors in the *DVF*. This is done based on distinct peaks in histograms of the *DVF*, allowing to derive the local average velocity vector. The latter can then be used as a replacement for unphysical results in the *DVF*. The relevance of the correction was discussed using the example of $SO_2$-emission-rate retrievals from UV camera data. Specifically, the $SO_2$ emissions of Mt. Etna (Italy) and Guallatiri (Chile) were investigated using two short example datasets (of about 15 minutes each). The gas velocities were analysed using the Farnebäck OF algorithm. Based

on these data we find, that unphysical motion vectors occur rather frequently and hence, often induce significantly underestimated $SO_2$-emission-rates. We further show, that the correction provides an efficient solution to this problem, resulting in more reliable velocity estimates and hence, in more robust $SO_2$-emission-rate retrievals.

The proposed method therefore provides an important and useful extension for OF based gas velocity retrievals.





## 6 Data and code availability

The analysis of the UV camera data was performed using the software *Pyplis* v0.12.0 (Gliß et al., 2017). The Etna data corresponds to the example dataset of *Pyplis* and can be downloaded from the website. The Guallatiri data, as well as the analysis scripts for both datasets can be provided upon request.

## 5 Appendix A: Detailed description of histogram analysis

The proposed histogram analysis of local optical flow *DVF*'s includes the following steps.

### A1 Retrieval of local displacement parameters

S1 Extraction of all displacement vectors $\boldsymbol{f} = [\Delta i, \Delta j]^T$ within specified ROI: $\boldsymbol{F} = \{\boldsymbol{f} \,|\, \boldsymbol{f} \,\forall (i,j) \in \mathrm{ROI}\}$, where $i, j$ denotes pixel coordinates in the image. Note that the considered pixels in the ROI may be further restricted, for instance by applying an intensity threshold (e.g. we use a $\tau_{\mathrm{on}}$ threshold to identify plume pixels).

S2 Determination of magnitude $|\boldsymbol{f}| = \sqrt{\Delta i^2 + \Delta j^2}$, and orientation angle $\varphi(\boldsymbol{f}) = \mathrm{atan2}(\Delta i, \Delta j)$ of all vectors in $\boldsymbol{F}$.

S3 Extraction of all vectors in $\boldsymbol{F}$ exceeding a certain magnitude threshold $|\boldsymbol{f}|_{\mathrm{min}}$: $\boldsymbol{F}' = \{\boldsymbol{f} : \boldsymbol{f} \in \boldsymbol{F} \wedge |\boldsymbol{f}| > |\boldsymbol{f}|_{\mathrm{min}}\}$.

S4 Calculation of orientation angle histogram $M_\varphi(\boldsymbol{F}', \Delta\varphi)$ of vectors in $\boldsymbol{F}'$ (with the histogram bin-width $\Delta\varphi$).

S5 Perform multi-peak analysis of $\mathcal{M}_\varphi$ using Multi-Gauss fit (for details see next Sect. A2).

S6 Use Multi-Gauss fit result to check, whether an unambiguous peak can be identified in $\mathcal{M}_\varphi$. If this is the case, estimate the expectation interval $\mathcal{I}_\varphi = [\varphi_\mu - n\varphi_\sigma, \varphi_\mu + n\varphi_\sigma]$ from 1. and 2. moment of $M_\varphi$ (with $n$ specifying a certain confidence level).

S7 Extraction of all flow vectors matching angular expectation interval $\mathcal{I}_\varphi$: $\boldsymbol{F}'' = \{\boldsymbol{f} : \boldsymbol{f} \in \boldsymbol{F}' \wedge \varphi(\boldsymbol{f}) \in \mathcal{I}_\varphi\}$.

S8 Calculation of displacement length histogram $M_{|\boldsymbol{f}|}(\boldsymbol{F}'', \Delta|\boldsymbol{f}|)$ from vectors in $\boldsymbol{F}''$ (with $\Delta|\boldsymbol{f}|$ being the bin-width in units of pixel displacements).

S9 Determine average displacement length $|\boldsymbol{f}|_\mu$ and standard deviation $|\boldsymbol{f}|_\sigma$ using first and second moment of $M_{|\boldsymbol{f}|}$.

### A2 Multi-Gauss fitting routine

The Multi-Gauss fitting routine is used to detect and parametrise distinct peaks in a given orientation histogram $M_\varphi$ calculated from a *DVF*. The parametrisation is performed by fitting a number $K$ of superimposed Gaussians of the form

$$f_K(\varphi; p) = \sum_{k=1}^{K} \mathcal{N}(\varphi; \mathcal{A}_k, \mu_k, \sigma_k) \tag{A1}$$





to $\mathcal{M}_\varphi$, with the Normal distribution

$$\mathcal{N}(\varphi; \mathcal{A}, \mu, \sigma) = \mathcal{A} \cdot e^{-\frac{(\varphi - \mu)^2}{2\sigma^2}} \qquad \text{(A2)}$$

and the corresponding parameter vector $p = (p_1, \ldots, p_K) = ((\mathcal{A}_1, \mu_1, \sigma_1), \ldots, (\mathcal{A}_K, \mu_K, \sigma_K))$. In order to achieve a physically more reliable result in the optimisation, we recommend to restrict the individual $p_k$ to certain expectation ranges, for instance:

- Minimum required amplitude: $\mathcal{A}_k > \mathcal{A}_{k,\min}$ (e.g. to avoid fitting all noise peaks)

- Lower threshold for standard deviation: $\sigma_k > \frac{\Delta\varphi}{2\sqrt{2\ln 2}}$ (FWHM must equal or exceed histogram bin resolution)

- Peak position in angular range: $\mu_k \in \{-180, 180\}$

- Define upper limit for allowed number of superimposed Gaussians: $K \leq K_{\max}$

A routine to perform this fit was written in Python and is implemented in the software package *Pyplis* (class *MultiGaussFit*). The algorithm aims to find the minimum number $K$ of Gaussians required to meet the constraint $\mathcal{C} : \mathcal{R}_{pp}(i) \leq \mathcal{A}_{k,\min}$, where $\mathcal{R}_{pp}(i)$ is the peak-to-peak value of the current fit residual $\mathcal{R}(i) = f_K(\varphi, p) - M_\varphi$ at iteration step $i$. If no additional peaks are found in $\mathcal{R}(i)$, the latest optimised parameter vector $p$ is assumed sufficient. Else, $p$ is extended by all additionally detected peaks (in $\mathcal{R}(i)$) and the least-squares fit is re-applied until the optimisation constraint $\mathcal{C}$ is fulfilled, or until a break constraint is met (e.g. maximum iteration reached, or maximum number of allowed Gaussians). An exemplary fit result is shown in Fig. 12.

**A3    Retrieval of main peak parameters from Multi-Gauss fit result**

The Multi-Gauss parametrisation of $\mathcal{M}_\varphi$ allows to identify the most prominent peak in the distribution (which may be a superposition of several Normals) and separate it from potential additional peaks. The latter can have significant impacts on the retrieved statistical parameters $\varphi_\mu, \varphi_\sigma$ (cf. Fig. 12).

Numerically, the retrieval of the main peak parameters from a given fit result vector $p$ is performed as follows:

1. For a given Gaussian $p_k$ in $p$, find all fitted Gaussians within a specified confidence interval $(n\sigma)_k$ around $p_k$ and calculate the integral value $I_k$ of the local overlap ($I_k$ corresponds to the number of vectors belonging to the main peak).

2. Do step 1. for all detected Gaussians in $p$, resulting in a vector $I_K$ of length $K$ containing integral values of the local overlaps.

3. Find the main peak position based on the index $k^*$ showing the largest integral value (in $I_K$)

4. Retrieve main peak parameters by calculating first and second moment of the corresponding local overlap $p_{k^*} : p' = \{p_k : \mu(p_k) \in [\varphi_{\min}, \varphi_{\max}]\}$ with $\varphi_{\min}, \varphi_{\max} = \mu_{k^*} \pm (n\sigma)_{k^*}$ (e.g. the two overlapping peaks at index $-74$ in Fig. 12).





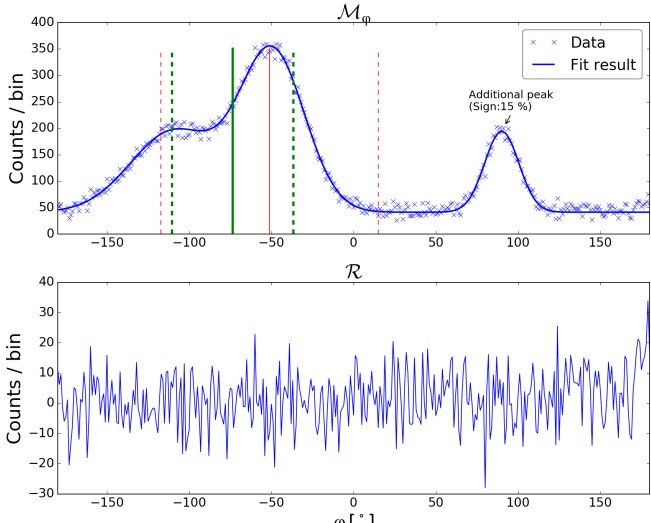

**Figure 12.** Top: Example fit result (blue line) of a multi-peak analysis applied to synthetic data (blue crosses). Bottom: corresponding fit residual. The dataset consists of three Gaussian Normals including Gaussian noise (with $\sigma_{\text{noise}} = 9$ counts). Two overlapping peaks are located at $\mu = -110\,(\mathcal{A} = 150, \sigma = 25)$ and $\mu = -50\,(\mathcal{A} = 300, \sigma = 20)$ (forming the predominant peak) and one additional peak at $\mu = 90\,(\mathcal{A} = 150, \sigma = 10)$ with a significance of 16%. Expectation parameters $(\varphi_\mu, \varphi_\sigma)$ are indicated with solid and dashed vertical lines, respectively. The latter were retrieved as described in Appendix A3, both including and excluding the additionally detected peak at $\mu = 90$, in red and green colours, respectively.

5. Retrieve mean and standard deviation of $p'$ based on the first and second moment of the resulting main peak distribution $f_{K'}(\varphi; p')$ (cf. Eq. A1), i.e.:

$$\varphi_\mu \;=\; \int\limits_{-\pi}^{\pi} \varphi\, f_{K'}(\varphi; p')\, d\varphi \tag{A3}$$

$$\varphi_\sigma \;=\; \sqrt{\int\limits_{-\pi}^{\pi} (\varphi - \mu_\varphi)^2\, f_{K'}(\varphi; p')\, d\varphi} \tag{A4}$$

5    resulting in an estimate of the predominant displacement direction in the ROI:

$$\varphi_{\text{glob}}(\text{ROI}) = (\varphi_\mu \pm \varphi_\sigma) \tag{A5}$$





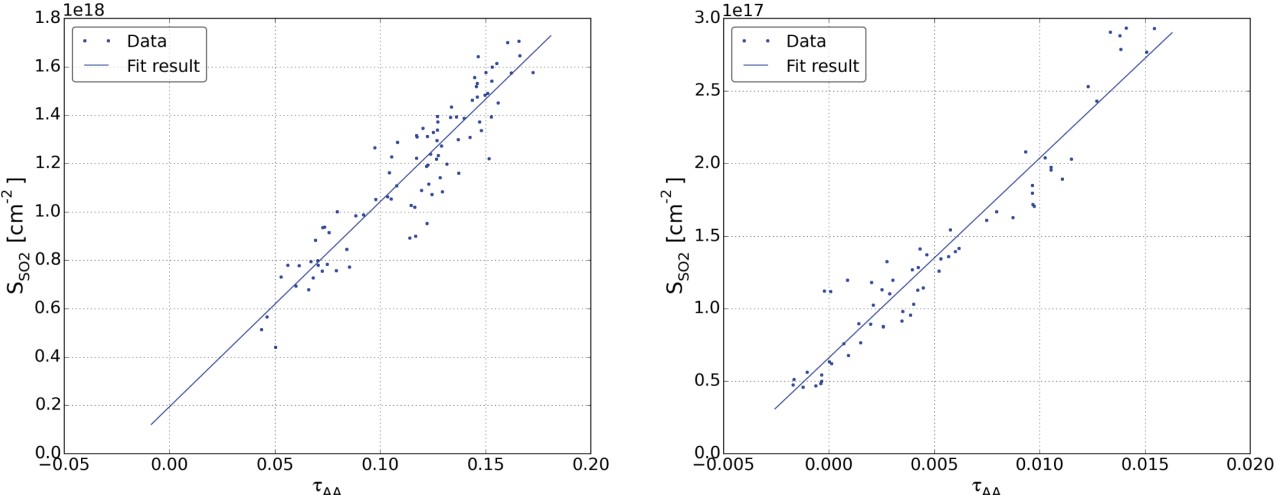

**Figure 13.** DOAS calibration curves of Etna data (left) and Guallatiri data (right) retrieved from AA images (not dilution corrected) within the image region covered by the DOAS-FOV (cf. Figs. 1 and 2). The corresponding $SO_2$-CDs retrieved with the DOAS instrument are plotted on the y-axis. Y-axis offsets were corrected for during the calibration of the AA images.

## Appendix B: Emission rate analysis supplementary information

### B1 DOAS $SO_2$ retrieval

The DOAS spectra from both datasets (Etna, Guallatiri) were analysed using the software DOASIS (Kraus, 2006). All spectra were corrected for electronic offset and dark current and were analysed using a clear sky Fraunhofer reference spectrum (FRS)

recorded close in time (to keep potential $O_3$ interferences at a minimum). In addition, a Ring spectrum, determined from the FRS, was fitted as well as the absorption cross sections (XS) of $SO_2$ (Hermans et al., 2009) and $O_3$ (Burrows et al., 1999). The latter were convolved with the instrumental line-spread-function (using the measured $334.15\,\mathrm{nm}$ mercury line). FRS and Ring were linked to each other and were allowed a slight shift of 0.2 nm and squeeze of 2%. The same shift and squeeze was allowed for the two XS, which were also linked. The retrieval was performed between $314.6 - 326.4\,\mathrm{nm}$. A third order DOAS

polynomial was fitted to account for broadband extinction and an additional zero order offset polynomial (fitted in intensity space) was included to account for instrumental effects (e.g. stray light).

### B2 Camera calibration

Fig. 13 shows the DOAS calibration curves retrieved for both datasets. The camera AA values correspond to the pixels covered by the DOAS-FOV shown in Figs. 1 and 2. Prior to the calibration, the camera images and the DOAS data were merged in

time.





## B3 Settings for optical flow retrieval

All relevant settings for the optical flow based gas velocity retrievals are summarised in Tab. 2

**Table 2.** Applied settings for Farnebäck algorithm (OpenCV implementation) and relevant parameters for the histogram post analysis.

| | Parameter | Etna | Gua | Description |
|---|---|---|---|---|
| **Farneback** | pyr_scale | 0.5 | 0.5 | Multi-scale analysis: downscale factor |
| | levels | 4 | 4 | Multi-scale analysis: pyramid levels |
| | winsize | 20 | 20 | Size of (gaussian) averaging neighbourhood |
| | iterations | 5 | 5 | Number of iterations |
| | poly_n | 5 | 5 | Size of neighbourhood for polynomial expansion |
| | poly_sigma | 1.1 | 1.1 | Standard deviation of smoothing kernel for poly. exp. |
| **Histo analysis** | $\Delta\varphi$ [°] | 15 | 20 | Bin width of $\mathcal{M}_\varphi$ |
| | $\Delta\|\boldsymbol{f}\|$ [pix] | 1 | 1 | Bin width of $\mathcal{M}_{\|\boldsymbol{f}\|}$ |
| | $\|\boldsymbol{f}\|_{\min}$ [pix] | 1.5 | 1.5 | Required minimum magnitude |
| | $n\sigma$ | 3 | 3 | Confidence level for retrieval of $\mathcal{I}_{\|\boldsymbol{f}\|}$ |
| | $\tau_{\min}$ | 0.15 | 0.02 | $\tau$ threshold for identifying plume pixels |
| | $r_{\min}$ | 0.1 | 0.1 | See Sect. 2.4.1 |
| | $\mathcal{S}$ | 0.2 | 0.2 | See Sect. 2.4.1 |

## B4 Results velocity cross-correlation

### B4.1 Etna

5  Cross correlation based gas velocities were retrieved for each of the two PCS lines, using two additional lines shifted by 40 pixels in the normal direction (cf. Fig. 1). Velocities of $v_{\mathrm{glob}} = 4.14\,\mathrm{m/s}$ and $v_{\mathrm{glob}} = 4.55\,\mathrm{m/s}$ could be retrieved for the young (orange) and aged (blue) plume, respectively. The results of the cross-correlation analysis (ICA time series) are shown in Fig. 14 in Appendix B4.

### B4.2 Guallatiri

10  Fig. 15 shows the result of the velocity cross-correlation analysis using the blue (*fumarole*) PCS retrieval line (cf. Sect. 3.2) resulting in a gas velocity of $v_{\mathrm{glob}} = 3.49\,\mathrm{m/s}$. The same velocity was assumed for the central crater plume (orange retrieval line in Fig. 2), where the cross-correlation algorithm did not succeed.





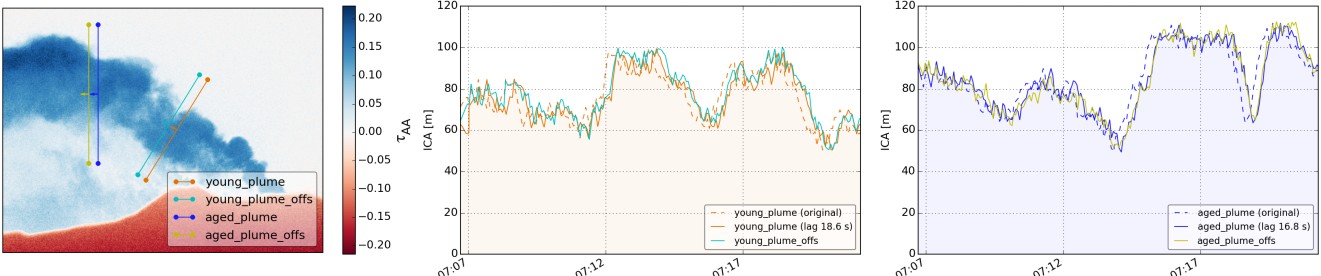

**Figure 14.** Result of velocity cross-correlation analysis for the Etna data. Left: Example plume AA image including two PCS lines (orange / blue) and corresponding offset lines (cyan / yellow) used for the analysis. Middle, right: corresponding ICA signals (same color codes) of the two PCS lines (original: dashed, shifted using correlation lag: solid) and the ICA signal of the offset lines (solid).

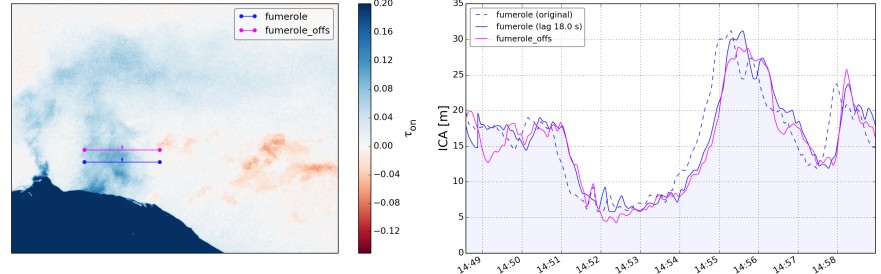

**Figure 15.** Result of velocity cross-correlation analysis for Guallatiri data (left), including the two PCS lines used for the retrieval. Right: corresponding ICA signals (dashed blue / magenta) and the shifted ICA signal from blue line, corresponding to the correlation lag of 18.0 s.

## B5 SO$_2$-emission-rate uncertainties

Uncertainties in the presented emission-rates (shaded areas in Figs. 6 and 10 top) were calculated based on Eq. 2 using Gaussian error propagation. Uncertainties in the plume distance (from uncertainty in plume azimuth and camera viewing direction), in the retrieved SO$_2$-CDs (from slope error the calibration polynomial) and in the effective gas velocities (Eq. 3) were considered. The latter were assumed constant for cross-correlation based velocities using $\Delta v_{\mathrm{glob}} = 1.5\,\mathrm{m/s}$. For the optical flow based retrievals, the uncertainties were estimated per image and PCS line as described in Sect. 2.4.1. Note, that uncertainties resulting from potential radiative transfer effects were not included. These are discussed in Sect. 2.3.3.

*Competing interests.* The authors declare that they have no conflict of interests.

*Acknowledgements.* We wish to thank the *Atmosphere and Remote Sensing* group from the Institute of Environmental Physics in Heidelberg, Germany, for support during the Etna field campaign in 2015. We acknowledge the support of F. Prata and H. Murray (Thomas) in planning



and conducting the measurements at Guallatiri. Useful discussions with T. Skauli and A. Donath related to the methodology are highly acknowledged. The work of K. Stebel and A. Kylling was partly supported by the European Research Council (ERC) under the European Union's Horizon 2020 research and innovation programme under grant agreement No 670462 (COMTESSA). Finally, J. Gliß acknowledges the Norwegian Institute for Air Research (NILU) and the Graduate Center of the University of Oslo (*UNIK*) for financial support.



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
