# Peer review of "Optical flow gas velocity analysis in plumes using UV cameras – Implications for SO2-emission-rate retrievals investigated at Mt. Etna, Italy, and Guallatiri, Chile"

_Atmospheric Measurement Techniques, 2017_

## Referee Comment (RC1) · C. Kern (Referee) · 9 Sep 2017

**General Remarks**

This manuscript describes an innovative method for improving optical flow velocity analysis in SO2 camera imagery of volcanic gas plumes. Optical flow algorithms cannot estimate velocity in regions of an image with little or no contrast. Such regions are often found in volcanic plume images, mostly in the center of the plume where detected gas column densities can be relatively homogeneous. Some existing optical flow methods

by default assign a very low velocity to low-contrast areas. The authors show that this issue can lead to underestimation of volcanic SO2 emission rates, as areas with high gas column density are assigned incorrect velocities.

Next, a method for improving velocity estimates in these low-contrast regions is described. A number of criteria are introduced to determine which pixels in a given image are associated with reliable velocity information. A histogram of velocities is derived from these reliable values, and areas in which a reliable velocity determination is not possible are filled in with the mean velocity obtained from this histogram.

Finally, the new method is tested on SO2 camera data from Mt. Etna (Italy) and Guallatiri (Chile). The authors make a compelling case for the improved quality of results obtained by their new technique when compared to other, previously employed methods. The manuscript is extremely well-written and presents relevant results. It will be particularly useful to the volcanic gas remote sensing community. I recommend that the manuscript be published in Atmospheric Measurement Techniques and below list only a few minor comments and corrections that might help improve it slightly.

Specific issues

Aside from some minor corrections, I really only have one issue with this study. The authors state that optical flow algorithms tend to fail for low-contrast image areas. It is true that all optical flow images rely on contrast in the image to derive velocity fields and cannot derive accurate velocities in low-contrast regions. However, this issue has been known for a long time. It's often referred to as the 'aperture problem' because, in general, contrast will depend on the size of the region that is being analyzed. Two adjacent pixels may have the same intensity, but by increasing the area of interest, one will eventually find an intensity gradient (unless the image is completely homogeneous in which case all is lost).

The method developed in this manuscript appears to work well. However, I wonder if there aren't other existing optical flow algorithms that essentially do the same thing. For

example, the original Horn-Schunck method introduces a global smoothness constraint to solve the aperture problem. The velocity field is determined in regions with sufficient contrast. Velocities in areas with low-contrast are determined by forcing the vector field to be smooth. If properly initialized, I would imagine that the Horn-Schunck method could give very good results even in plume areas with low contrast, and even when the Farnebaeck method fails.

I don't doubt that the method presented by the authors here works well, but I do think it could be worth reviewing existing optical flow methods in a bit more detail to see if there aren't already some that essentially solve the problem in the same or a similar way. In particular, a discussion of the differences between the author's method and the original Horn-Schunck method could be warranted.

Minor corrections

Title: You might consider changing the title slightly to reflect the fact that the main focus of this manuscript is the improvement of the velocity analysis techniques. Also, since the techniques presented here apply equally well to UV and IR SO2 cameras, I would suggest replacing 'UV cameras' with 'SO2 cameras' in the title. Perhaps: Improved optical flow velocity analysis in SO2 camera images of volcanic plumes – Implications for SO2 emission rate retrievals investigated at Mt Etna, Italy and Guallatiri, Chile

Introduction: The phrases 'for example', 'for instance', and 'e.g.' are used overwhelmingly often in the introduction. Consider rephrasing some of these sentences such that these phrases are not needed as often.

P2, L13 – Please clarify: CDs are not simply multiplied by the gas velocities. Instead, the product needs to be integrated across a cross-section of the plume.

P2, L19 – Please add a reference to the original work by McElhoe and Conner who, as it turns out, already built an SO2 camera in the 1980s but seldom receive credit for it: McElhoe, H. B., and W. D. Conner (1986), Remote Measurement of Sulfur Dioxide

[Figure]

Emissions Using an Ultraviolet Light Sensitive Video System, J. Air Pollut. Control Assoc., 36(1), 42–47, doi:10.1080/00022470.1986.10466043.

P2, L22 – Consider changing 'volcanic craters' to 'individual volcanic vents'

P2, L24 – Consider removing 'hence' before 'often'

P2, L34 – Consider removing 'often' before 'tend'

P2, L34 – Some OF algorithms have ways of dealing with low-contrast areas in images (see discussion above). Perhaps it's better to simply state that OF algorithms cannot track movement in the absence of intensity gradients, so information on velocities in these areas must be obtained from elsewhere.

P4, L9 – '... angle between plume DIRECTION and image plane'

Figure 1 caption - 'the town OF Milo'

Figure 1 - In my copy, the colors are not very clearly identifiable. The line that is supposedly yellow appears more green, and there are two blue lines. Perhaps choose a different color scheme, or use solid and dashed lines?

P5, L6- 'town OF Milo'.

P5, L13- 'downwind OF the source'

Figure 2 – Some of the structures in the Guallatiri plume look similar to structures that I have obtained when not recording images in both channels coincidently. That is, if the plume moves between the time that the on-band image is taken and when the off-band image is taken, this can lead to artefacts. Is this something you have observed before? How fast are the images taken in succession? Is this why you prefer to perform the optical flow analysis on the on-band image rather than on the optical depth (AA) image?

Figure 2 – Can you please clarify why the Guallatiri image appears to have only 300

horizontal pixels while the Etna image has 600 despite the fact that the same detector was used?

P7, L30 – Figure 13 shows the DOAS calibration curves. However, the caption says that the calibration was retrieved from AA images that were not corrected for the dilution effect. Why? The text on page 7 says that the AA images were corrected first. Please clarify.

P8, L11 – Please remove last sentence in this paragraph as it is repeated twice.

P8, L31 and P9, L13 and P11, L2 – Please replace 'constraint' with 'constrained'.

P9, L10 – '. . . vectors manifest as A constant offset and as A peak. . .'

P11, L27 – Can you please explain how you arrive at 15% uncertainty? Is this a guess?

P12, L17 - Suggest rewording this sentence for clarity: '. . . all pixels on I while Xpixok corresponds to the SO2 ICA considering only pixels showing reliable flow vectors.

P12, L21 – Is there a specific reason for performing the velocity analysis on the on-band images rather than on the AA images?

Figures 4 and 8– It's very difficult to see the vectors. Can the color scheme be improved, or the vectors made longer/bolder?

P17, L12 – I assume that the 'flow_histo' method should be the 'flow_hybrid' method.

P18, L3 – '. . . gradual decrease in the effective gas velocities after 14:54 at the main crater, which could. . . I wonder whether the velo: glob method should even be discussed here if it failed to retrieve velocities for the main crater. Perhaps omit it in Figure 10 (left) and only discuss it for the fumaroles?

Figure 11 – Similarly, it might be best to only plot the glob method for the fumaroles, not for the main crater.

P19, L15 – '. . . during the observation PERIOD.'

P20, L3 – Etna's (please add apostrophe)

P20, L6 – 'In the case of Guallatiri, these are the first SO2 emission rates reported in the literature.'

Figures 1, 2, and 13 – The unit for column density is molecules/cm2. Adding the 'molecules' in plot labels would improve clarity.

Figures 14 and 15 – Is there a reason why the AA is plotted in figure 14, but only the on-band optical depth is plotted in figure 15? Perhaps it's best to be consistent?

---

## Referee Comment (RC2) · Anonymous Referee #3 · 26 Oct 2017

**General comments:**

This paper introduces a new post-correction method for the output of the Farneback optical flow algorithm. The Farneback algorithm tends to fail in low contrast regions, such as the central plume region, and apparently does not give any uncertainty estimate nor quality flags for its output that would allow a straight-forward filtering of unreliable motion vectors. The unphysical output is characterized by short randomly oriented vectors. The proposed post-correction method exploits this feature and replaces the unphysical output within a selected plume region by a local average obtained from a

histogram analysis. Better velocity estimates for the plume central regions are thus obtained and the corresponding estimates of sulfur dioxide emission rates improved.

Evidence on the performance of the proposed post-correction algorithm is given for two different volcano cases: Mt. Etna in Italy and Guallatiri volcano in Chile. The algorithm has many tunable parameters but appears to perform well with the selected parameters for the two cases. The results are comparable to two reference methods: a cross-correlation method and using the uncorrected output from the Farneback method. The post-corrected emission rates are larger than the uncorrected ones, as expected when randomly oriented vectors are replaced by the local average. For the remotely located Guallatiri, estimates of the sulfur dioxide emission rate are given for the first time.

The manuscript is within the aims and scope of AMT and is suitable for publication with the corrections listed below. The scientific methods are commonly used in the field and presentation quality is generally good.

**Specific comments:**

P7, L8-11: Could you please add more details on the determination of the background intensities. Do you fit functions to the image background and then evaluate the fitted functions at the plume location to get the background intensity?

P10, Fig.3: Filtering out the vectors shorter than 1.5 pixels seems to effectively remove the unphysical vectors in the background. Considering the vectors in the remaining plume (red / blue shaded areas) how do you distinguish vectors due to real turbulent motion from the unphysical output of the Farneback algorithm?

P12, L17: Please write out the acronym ICA as it is introduced here for the first time.

P15, Fig. 7 and P19, Fig. 11: y-axis labels, legends and caption texts are inconsistent in describing what is plotted: absolute deviation $\Delta\Phi$, relative deviation $\Delta\Phi/\Phi_{hybrid}$ or ratio $\Phi/\Phi_{hybrid}$. The legend appears to be correct, i.e. ratios are plotted. Please make these consistent.

P19, L5: There is no evidence in the paper to make a generic statement about OF algorithms. Only the Farneback algorithm was applied to two cases. Please be more specific, for example: "We showed that the Farneback algorithm is able to resolve ..."

P25, Table 2: How sensitive is your method to the selection of parameters for the Farneback algorithm and the histogram analysis given here, and also for the selection of the ROI? Did you perform any sensitivity analysis? Is it just trial and error to arrive at the given values for the parameters? Why are both $\tau_{min}$ and $|f|_{min}$ needed in the histogram analysis, as Fig. 3 suggests that $|f|_{min}$ alone selects the plume region?

**Suggestions for technical improvements:**
P4, L15 and throughout the text: please do not abbreviate the word "Table" (see Manuscript preparation guidelines).
P6, Fig. 2: there is no label (b). Please remove the "in (b)" at the end of the caption or label the panels as (a) and (b).
P6, L5: replace "data is" with "data are".
P8, L11 and throughout the text: please use "Figure" at the beginning of a new sentence, not "Fig.".
P17, L12: replace "flow_histo" with "flow_hybrid"
P20, L3: replace "Etnas" with "Etna's"

––––––––––––––––––––

---

## Author Comment (AC1) · 30 Nov 2017

Dear Dr. Joiner,

We like to thank the two referees for their very helpful comments which helped us to significantly improve the manuscript. Below we repeat the comments from the two referees and our corresponding responses (in blue) and where applicable, we also include the changes that were applied to the manuscript (also in blue, within quotation marks, new text in *italic*). If not specified otherwise, all references to sections of the text (i.e. page, line numbers) refer to the originally submitted manuscript. Before addressing the referees' comments, we list all additional changes that were applied in the revised manuscript.

**Additional changes applied to the manuscript**

- Replaced all occurrences of "SO2-emission-rates" with "SO2 emission-rates".
- Replaced all occurrences of "time-series" with "time series".
- **P.2, L. 9:** removed following part in sentence: "due to the variety of different technologies,".
- P.2, L. 14: removed following part in sentence: "if multiple scanners are available" and added reference to the paper of Johansson et al. 2009: The dual-beam mini-DOAS technique—measurements of volcanic gas emission, plume height and plume speed with a single instrument, Bulletin of Volcanology, 71, 747–751, doi:10.1007/s00445-008-0260-8.
- **P.2, L. 18:** Replaced "Another type of available instrumentation" with "A more recent measurement technique".
- P.3, L. 22 and P.8, L. 18: Changed "aerosole" to "aerosol".
- **P.7, L. 26:** Corrected reference to figure in paper of Gliß, et al., 2017 (Pyplis software) from "Fig. 12" to "*Fig. 10*".
- **P.8, L. 31:** Added reference by changing "(e.g. in low-contrast plume regions)" to "(e.g. in low-contrast plume regions, cf. Sect. 1)".
- P.11, L. 21f: The text was shortened, since the information about the benchmarks is now already provided in the revised Introduction (see below). Before:

"It is therefore recommended, to assess the general performance of the used OF algorithm independently and before applying the histogram correction. For the latter, optical flow inter-comparison benchmarks (e.g. Baker et al., 2011, Menze and Geiger, 2015) can provide useful information to assess the performance (e.g. accuracy) and applicability (e.g. computational demands) of individual OF algorithms." After:

"It is therefore recommended, to assess the performance of the used OF algorithm independently and before applying the histogram correction (e.g. Baker et al., 2011, Menze and Geiger, 2015)."

- **P.14, L. 9**: Replaced "... and may thus, not to be misinterpreted ..." with "... and are thus, not to be misinterpreted ...".
- **P. 16, L. 6:** corrected typo: "fumerolic" → "fumarolic".
- **P. 16, L. 11 & L. 13:** corrected typo (2x): "colors" → "colours".
- P. 12, L. 20: Renamed section 3.1 from "Etna" to "Etna results".
- P. 20, L. 10: Replaced "... of the volcano" with "... of Guallatiri".

- **Figure 2 (left):** Updated colour style of camera viewing and plume direction vectors in map (see also comment below from reviewer 1 related to colours in Figure 1).
- Figure 6, caption: Wrong reference to figure. Replaced "shown in Fig. 2" with "shown in Fig. 1".
- **Figure 14:** Updated colours of retrieval lines (more details in the comments of Reviewer 1 below) and updated the caption text which now reads as: *"Result of velocity cross-correlation analysis for the Etna data. Left: Example plume AA image including two PCS lines (orange / blue) and corresponding offset lines (green / red) used for the analysis. Middle, right: corresponding time-series of the integrated AA values along the two PCS lines (original: dashed, shifted using correlation lag: solid) and along the offset lines (solid) using the same colour scheme."*
- Figure 15: Corrected typo "fumerole" to "fumaroles" in legends of subplots. Right plot: changed y-axis label from "ICA [m]" to "Integrated τon [m]". Slightly updated the caption text which now includes a hint that here, the on-band ODs are used rather than the AA values as in Fig. 14. The caption now reads as:
   "Result of velocity cross-correlation analysis for Guallatiri data. Left: Example plume on-band OD image (τon) including the PCS (blue) and offset line (magenta) used for the analysis. Right: corresponding time-series of integrated on-band ODs (dashed blue / magenta) and further, the shifted PCS signal corresponding to the retrieved correlation lag of 18.0 s (solid blue). Note that here, the velocity analysis was applied using a time-series of on-band OD images rather than the τAA which was used in Fig. 14."
- Status update of reference to paper of Gliß et al., 2017 (Pyplis software) from "Gliß, J., Stebel, K., Kylling, A., Dinger, A. S., Sihler, H., and Sudbø, A.: Pyplis A Python software tool box for the analysis of UV camera data to study SO2 plumes, submitted for publication in Computers and Geosciences (Elsevier), 2017."
   to

"Gliß, J., Stebel, K., Kylling, A., Dinger, A., Sihler, H., and Sudbø, A.: Pyplis - A Python Software Toolbox for the Analysis of SO2 Camera Data, Preprints, doi:10.20944/preprints201710.0085.v1, under review for publication in journal Geosciences (MDPI), 2017."

• Added the following sentence to the Acknowledgements: "Finally, we wish to thank the reviewer C. Kern and one anonymous reviewer as well as J. Joiner as Editor for their very valuable comments and suggestions as well as support during the review phase."

**Review 1 (Reviewer: C. Kern)**

Interactive comment on "Optical flow gas velocity analysis in plumes using UV cameras – Implications for SO2-emission-rate retrievals investigated at Mt. Etna, Italy, and Guallatiri, Chile"

C. Kern (Referee) ckern@usgs.gov

Received and published: 9 September 2017

**General Remarks**

This manuscript describes an innovative method for improving optical flow velocity analysis in SO2 camera imagery of volcanic gas plumes. Optical flow algorithms cannot estimate velocity in regions of an image with little or no contrast. Such regions are often found in volcanic plume images, mostly in the center of the plume where detected gas column densities can be relatively homogeneous. Some existing optical flow methods by default assign a very low velocity to low-contrast areas. The authors show that this issue can lead to underestimation of volcanic SO2 emission rates, as areas with high gas column density are assigned incorrect velocities.

Next, a method for improving velocity estimates in these low-contrast regions is described. A number of criteria are introduced to determine which pixels in a given image are associated with reliable velocity information. A histogram of velocities is derived from these reliable values, and areas in which a reliable velocity determination is not possible are filled in with the mean velocity obtained from this histogram.

Finally, the new method is tested on SO2 camera data from Mt. Etna (Italy) and Guallatiri (Chile). The authors make a compelling case for the improved quality of results obtained by their new technique when compared to other, previously employed methods. The manuscript is extremely well-written and presents relevant results. It will be particularly useful to the volcanic gas remote sensing community. I recommend that the manuscript be published in Atmospheric Measurement Techniques and below list only a few minor comments and corrections that might help improve it slightly.

**Specific issues**

Aside from some minor corrections, I really only have one issue with this study. The authors state that optical flow algorithms tend to fail for low-contrast image areas. It is true that all optical flow images rely on contrast in the image to derive velocity fields and cannot derive accurate velocities in low-contrast regions. However, this issue has been known for a long time. It's often referred to as the 'aperture problem' because, in general, contrast will depend on the size of the region that is being analyzed. Two adjacent pixels may have the same intensity, but by increasing the area of interest, one will eventually find an intensity gradient (unless the image is completely homogeneous in which case all is lost). The method developed in this manuscript appears to work well. However, I wonder if there aren't other existing optical flow algorithms that essentially do the same thing. For example,

the original Horn-Schunck method introduces a global smoothness constraint to solve the aperture problem. The velocity field is determined in regions with sufficient contrast. Velocities in areas with low-contrast are determined by forcing the vector field to be smooth. If properly initialized, I would imagine that the Horn-Schunck method could give very good results even in plume areas with low contrast, and even when the Farnebaeck method fails.

I don't doubt that the method presented by the authors here works well, but I do think it could be worth reviewing existing optical flow methods in a bit more detail to see if there aren't already some that essentially solve the problem in the same or a similar way. In particular, a discussion of the differences between the author's method and the original Horn-Schunck method could be warranted.

It is true, and we absolutely agree, that there might be optical flow (OF) algorithms that show a better performance in extended homogeneous image areas than the Farnebäck algorithm, which is used in this paper to illustrate the new method. As discussed in the revised version of the manuscript, some OF algorithms, such as the quoted Horn-Schunck algorithm, use *global* smoothness constraints to overcome the aperture problem and it is true, that such methods are typically better in *filling-in* the optical flow in homogeneous image regions. However, it is not the purpose, and beyond the scope of the paper to review the vast literature on available OF algorithms and to provide a suggestion of a best candidate for the discussed application of plume imaging systems. Especially because the latter also depends on factors such as the availability of the source code of a certain OF algorithm (e.g. dependent on the used framework such as MATLAB, OpenCV, etc.), or the computational efficiency (e.g. for near real-time analysis), where *local* methods, such as the used Farnebäck algorithm, typically show better performance (e.g. Fleet and Weiss, 2006, reference details can be found in the revised version of the manuscript). For these very reasons, we developed the proposed method and the main idea is, that it is decoupled from a specific OF implementation and can therefore be applied to the output of any OF algorithm. This gives the user the freedom to choose the best suitable OF algorithm for their data situation (e.g. considering also factors such as noise-performance or computation time). In the best case (i.e. OF algorithm performs well within the considered ROI), our method serves as an external quality-check, allowing to rule out failures in the velocity analysis. For all other cases, our method can identify and correct for outliers as described in the manuscript.

Again, we remark that we completely agree with the reviewer's suggestions. Therefore, in order to highlight the differences between our *post-analysis* method and different spatial coherency constraints that are *intrinsically* implemented in optical flow algorithms and in different ways (i.e. *locally* or *globally*), we reformulated and extended the relevant paragraph in the introduction (P. 2, L. 29 – P. 3, L. 10).

**Changes to the manuscript (P. 2, L. 29 – P.4, L. 6 in revised manuscript):**

"OF algorithms can detect motion at the pixel-level by tracking distinct image features in consecutive frames. In the following, the basic principles of the OF computation are briefly introduced, as well as different optimisation strategies (see e.g. Jähne, 1997, Fleet and Weiss, 2006, Fortun et al., 2015 for a comprehensive introduction into the topic). OF algorithms are based on the assumption that a certain image quantity, such as the brightness I or the local phase  $\phi$ , is conserved between consecutive frames. Then, a continuity equation of the form

**$\partial_t g + \mathbf{f} \nabla_{ij} g = 0$**

can be used to describe the apparent motion of brightness (or phase) patterns between two frames. Here,  $f = [u, v]^T$  denotes the flow vector in the detector coordinate system i, j. g is the conserved quantity (e.g.  $I, \phi$ ),  $\nabla_{ij} = [\partial_i, \partial_j]^T$  and  $\partial_t$  denote the spatial and temporal differentiation operators. Eq. 1 is typically referred to as the optical flow constraint (OFC) equation and can be solved numerically per image-pixel, for example, using a least-squares or a total least-squares optimisation scheme. The OFC states an ill-posed problem, as it seeks to find the two velocity components u and v from a single constraint (i.e. I or  $\phi$ , cf. Eq. 1). This is commonly referred to as the aperture problem and is typically accounted for by introducing further constraints that impose spatial coherency to the flow field. These can be subdivided into local and global constraints, or a combination of both (e.g. Bruhn et al., 2005). Local methods (e.g. Lucas and Kanade, 1981) apply the coherency constraint only within a certain neighbourhood around each pixel (the size of this aperture can usually be set by the user). Thus, for pixel-positions that do not contain at least one trackable feature within the neighbourhood specified by the aperture size, the algorithm will fail to detect motion. We shall see below, that this can be a fundamental problem for the emission-rate analysis using plume imagery, in case extended homogeneous plume regions coincide with a retrieval transect  $\ell$ . The problem is less pronounced for OF algorithms using global constraints (e.g. the algorithm by Horn and Schunck, 1981 which is used in Kern et al., 2015), which can propagate reliable motion vectors over larger image areas. However, note that, dependent on the optimisation strategy, global regularisers are often more sensitive to noise (e.g. Barron et al., 1994) and are typically computationally more demanding (e.g. Fleet and Weiss, 2006).

Most of the modern OF algorithms include a multi-scale analysis where the flow-field is retrieved from coarse to fine features, using image pyramids combined with suitable warping techniques (e.g. Anandan, 1989). This can significantly increase the robustness of the results and is of particular relevance in case of large displacements (i.e. several imagepixels, e.g. Beauchemin and Barron, 1995, Fleet and Weiss, 2006).

Optical flow inter-comparison benchmarks (e.g. Baker et al., 2011, Menze and Geiger, 2015) can provide useful information to assess the performance (e.g. accuracy) and applicability (e.g. computational demands, availability of source-code) of different OF algorithms. Particularly important for the emission-rate analysis is the computational efficiency as well as the performance within homogeneous image-regions. As discussed above, the latter may be optimised via the incorporated coherency constraints (e.g. by increasing the local averaging neighbourhood around each pixel) or by performing a multi-scale analysis. However, this can significantly increase in the required computation times (e.g. Fleet and Weiss, 2006) and may therefore be inapplicable, especially for near-real time analyses.

Given these challenges, in many cases the choice of a suitable OF algorithm will be a tradeoff between computational efficiency and the performance within homogeneous imageregions. In order to rule out potential failures in the OF retrieval, it is therefore highly desirable to assess the OF performance before calculating the emission-rates. In this paper, we propose a new method, which analyses an OF displacement-vector-field (DVF) in order to identify and correct for potentially unphysical OF motion-estimates. The correction is performed in a localised manner, within a specific region-of-interest (ROI) in the images (e.g. in proximity to a plume transect  $\ell$ ). It measures the local-average-velocity-vector (LAVV) within the ROI, based on distinct peaks in histograms computed from the local DVF. The strengths of the method are 1) that it is independent of the choice of the OF algorithm and 2) that the additional computational demands are small compared to the OF computation time. The new method is introduced using the Farnebäck optical flow algorithm (Farnebäck, 2003) which showed promising results in Peters et al. (2015) and which is freely available in the OpenCV library (e.g. Bradski, 2000). We use two different volcanic datasets recorded at Mt. Etna, Italy and Guallatiri, Chile to show, that our method can successfully detect and correct for unphysical OF motion estimates during the emission-rate analysis."

**Minor corrections**

Title: You might consider changing the title slightly to reflect the fact that the main focus of this manuscript is the improvement of the velocity analysis techniques. Also, since the techniques presented here apply equally well to UV and IR SO2 cameras, I would suggest replacing 'UV cameras' with 'SO2 cameras' in the title. Perhaps: Improved optical flow velocity analysis in SO2 camera images of volcanic plumes – Implications for SO2 emission rate retrievals investigated at Mt Etna, Italy and Guallatiri, Chile.

We agree and renamed the title of the manuscript as follows:

**Changes to the manuscript:**

"Improved optical flow velocity analysis in SO2 camera images of volcanic plumes -Implications for emission-rate retrievals investigated at Mt. Etna, Italy and Guallatiri, Chile"

Introduction: The phrases 'for example', 'for instance', and 'e.g.' are used overwhelmingly often in the introduction. Consider rephrasing some of these sentences such that these phrases are not needed as often.

We agree and rephrased relevant parts of the introduction.

P2, L13 – Please clarify: CDs are not simply multiplied by the gas velocities. Instead, the product needs to be integrated across a cross-section of the plume.

We agree and rephrased this sentence as follows:

**Changes to the manuscript:**

"Gas-emission-rates (or fluxes) of the sources are typically retrieved along a plume transect by integrating the product of the measured CDs with the local gas velocities in the plume."

P2, L19 – Please add a reference to the original work by McElhoe and Conner who, as it turns out, already built an SO2 camera in the 1980s but seldom receive credit for it:
McElhoe, H. B., and W. D. Conner (1986), Remote Measurement of Sulfur Dioxide Emissions Using an Ultraviolet Light Sensitive Video System, J. Air Pollut. Control Assoc., 36(1), 42–47, doi:10.1080/00022470.1986.10466043.

We followed the suggestion and included the reference.

P2, L22 – Consider changing 'volcanic craters' to 'individual volcanic vents'

We removed the example completely in the revised version of the manuscript.

P2, L24 - Consider removing 'hence' before 'often'

We followed this suggestion.

P2, L34 – Consider removing 'often' before 'tend'

This was resolved in the reformulated paragraph of the introduction (discussed above).

P2, L34 – Some OF algorithms have ways of dealing with low-contrast areas in images (see discussion above). Perhaps it's better to simply state that OF algorithms cannot track movement in the absence of intensity gradients, so information on velocities in these areas must be obtained from elsewhere.

This was resolved in the reformulated paragraph of the introduction (discussed above).

P4, L9 – '... angle between plume DIRECTION and image plane'

We followed the suggestion.

Figure 1 caption - 'the town OF Milo'

We followed the suggestion.

Figure 1 - In my copy, the colors are not very clearly identifiable. The line that is supposedly yellow appears more green, and there are two blue lines. Perhaps choose a different color scheme, or use solid and dashed lines?

We followed this suggestion and updated the colours of the retrieval lines in the displayed plume image (right) from yellow  $\rightarrow$  red and cyan (light blue)  $\rightarrow$  green. We also changed the colours of the viewing and plume vectors in the map (left).

**Changes to the manuscript**

- Colours of lines in Fig. 1:
- Caption of Fig. 1 now reads as:

"Left: Etna overview map showing position and viewing direction of the camera (camera cfov, fov) which was located on a roof-top in the town of Milo. Also indicated is the summit area (source) and the plume azimuth (plume direction). Right: example SO2-CD image of the Etna plume including two PCS lines (orange / blue) used for emission-rate retrievals and two corresponding offset lines (green, red), that are used for cross-correlation based plume velocity retrievals (cf. Appendix B4). Position and extent of the DOAS-FOV for the camera calibration is indicated by a green spot. Note that the displayed plume image is size reduced by a factor of two (Gauss pyramid level 1)."

- The colour changes were applied to Figs. 6, 7 and 14, accordingly.
- All occurrences / mentioning of the colours in the text and in the captions of the mentioned Figures were replaced accordingly.

P5, L6- 'town OF Milo'.

We followed the suggestion.

P5, L13- 'downwind OF the source'

We followed the suggestion.

Figure 2 – Some of the structures in the Guallatiri plume look similar to structures that I have obtained when not recording images in both channels coincidently. That is, if the plume moves between the time that the on-band image is taken and when the offband image is taken, this can lead to artefacts. Is this something you have observed before? How fast are the images taken in succession? Is this why you prefer to perform the optical flow analysis on the on-band image rather than on the optical depth (AA) image?

Indeed, these structures are due to the time-shift between the on and off-band images and are likely arising from the highly turbulent conditions during the measurements. The average time shift between the on and corresponding off-band images in the time-series is  $2.76 \pm 0.06 \ s$ . In case of the Etna plume, the time difference is smaller and amounts to  $1.85 \pm 0.06 \ s$  and here, the structures could not be observed (see also the supplementary videos, which show the differences more clearly). The effect is much more pronounced in the turbulent Guallatiri data which has a lower sampling rate. We therefore decided to retrieve the OF both for the Etna and for the Guallatiri data from the on-band images, rather than from the AA images. Furthermore, in the case of Guallatiri, the results of the cross-correlation velocity analysis (cf. Fig. 14) were considerably better when using the on-band ODs rather than the AA values (as in the Etna example, cf. Fig. 14). Therefore, in case of Guallatiri, we used the on-band OD images for the cross-correlation analysis.

Figure 2 – Can you please clarify why the Guallatiri image appears to have only 300 horizontal pixels while the Etna image has 600 despite the fact that the same detector was used?

The original detector size is 1344x1024 pixels. Both the Etna image and the Guallatiri image were reduced in size using a Gaussian pyramid (Etna: level 1, i.e. factor of 2, Guallatiri: level 2, i.e. factor of 4). In order to avoid confusion, we replotted the Guallatiri image at pyramid-level 1 and included a note in the caption (as well as in the caption of Fig. 1, see above).

**Changes to the manuscript (Figure 2):**

Replotted image (right) at pyramid level 1 and further, updated the colours in the overview map (left). Slightly reformulated the caption which now reads as: *"Left: Guallatiri overview map showing position and viewing direction of camera (camera*

cfov, fov), summit area (source) as well as the plume azimuth (plume direction). Right: example SO2-CD image of the Guallatiri emissions including two PCS lines used to retrieve SO2 emission-rates from the central crater (orange) and from a fumarolic field (blue) located behind the flank in the viewing direction. An additional line (magenta) is used to estimate gas velocities using a cross-correlation algorithm (relative to blue PCS line, cf. Appendix B4). Position and extent of the DOAS-FOV is indicated by a green spot. Note that the displayed plume image is size reduced by a factor of two (Gauss pyramid level 1)."

P7, L30 – Figure 13 shows the DOAS calibration curves. However, the caption says that the calibration was retrieved from AA images that were not corrected for the dilution effect. Why? The text on page 7 says that the AA images were corrected first. Please clarify.

We agree this aspect needs further explanation. Indeed, the displayed calibration curves correspond to the calibration data using the uncorrected AA images (AAuncorr). These are calibrated using the also uncorrected SO2-CDs measured with the DOAS instrument. In a next step, the camera images are corrected for the signal dilution so the recomputed AA values correspond to dilution corrected AAcorr values. In order to perform the calibration, we assume, that the fitted DOAS-CDs and the measured AA values include the same amount of light dilution and further, that the slope of the calibration curve remains linear up into the regime of the AAcorr values. We are convinced that this is justified for these data, since the plume aerosol optical densities as well as the SO2-CDs were low to moderate in both examples. Thus, we can assume that our data was not prone to complex radiative transfer leading to non-linear calibration curves (as in the case of your observations at Kilauea: Kern et al., 2013: "Applying UV cameras for SO2 detection to distant or optically thick volcanic plumes"). Based on this assumption of a linear calibration curve, we can extrapolate the fitted calibration polynomial to the AA value range of the corrected images, in order to retrieve the SO2-CDs for the dilution corrected AA values. In order to make this point clearer, we extended the discussion in Sect. 2.3.1 and Appendix B2 as follows:

**Changes to the manuscript**

• Sect. 2.3.1:

"The dilution corrected AA images were calibrated using the DOAS calibration curve shown in Appendix B2. The linear calibration polynomial was retrieved prior to the analysis using camera AA-values that were not corrected for the signal-dilution effect and the corresponding SO2-CDs measured with the DOAS spectrometer (for details see Appendix B2)."

• Appendix B2:

"Figure 13 shows the DOAS calibration curves retrieved for both datasets. The camera AA values correspond to the pixels covered by the DOAS-FOV shown in Figs. 1 and 2. Prior to the calibration, the camera images and the DOAS data were merged in time. Note that the calibration data displayed in Fig. 13 is not corrected for the signal-dilution effect. In order to calibrate the dilution corrected AA-images, the fitted calibration polynomial was extrapolated into the AA regime of the dilution corrected images. This is based on the assumption that the calibration curve remains linear also at larger optical densities, which is justified by the considerably good plume conditions (low to no condensation) and the low to moderate range of observed SO2-

CDs (cf. Sect. 2.3.3, see also 5 Kern et al. 2013). Furthermore, this calibration method assumes that the retrieved DOAS SO2-CDs exhibit approximately the same amount of signal dilution as the camera imagery. This is justified by the fact, that the DOAS analysis was applied in a wavelength range coinciding with the on / off-band regime of the camera filters (cf. previous Sect. B1)."

P8, L11 – Please remove last sentence in this paragraph as it is repeated twice.

We thank the reviewer for this hint and followed the suggestion.

P8, L31 and P9, L13 and P11, L2 – Please replace 'constraint' with 'constrained'.

We thank the reviewer for this hint and followed the suggestion. We also corrected two more occurrences of this typo in the "Abstract" and on P.12, L. 23.

P9, L10 – '... vectors manifest as A constant offset and as A peak. ..'

We followed the suggestion.

P11, L27 – Can you please explain how you arrive at 15% uncertainty? Is this a guess?

Indeed, this choice is somewhat arbitrary since it is beyond the scope of the study to verify the accuracy of the Farnebäck OF algorithm based on these data. We used the estimate of 15% based on the findings of Menze and Geiger, 2015 who find that 50% ( $\approx 1\sigma$ ) of all vectors are within a disparity radius of 5%. Based on this we assume, that nearly all vectors ( $\approx 3\sigma$ ) fall into a disparity radius of 15%. To make this a bit clearer, we reformulated the corresponding paragraph 2.4.1.:

**Changes to the manuscript**

P11, L.21f: "Please note that the method cannot account for any uncertainties intrinsic to the used OF algorithm since these directly propagate to the derived histogram parameters. It is therefore recommended, to assess the performance of the used OF algorithm independently and before applying the histogram correction (see e.g. Baker et al., 2011, Menze and Geiger, 2015). The Farnebäck algorithm used in this study showed sufficient performance both in Peters et al. (2015) and in the KITTI benchmark (cf. Menze and Geiger, 2015). The latter find that the algorithm yields correct velocity estimates in about 50% of all cases (approximately  $1\sigma$ ). Here, "correct" means, that the disparity between a retrieved flow-vector endpoint and its true value does not exceed a threshold of 5%. We therefore assume that the majority (i.e.  $\approx 3\sigma$ ) of all successfully constrained flow vectors lie within a disparity radius of 15%. Based on this, we assume an intrinsic, conservative uncertainty of 15% for the effective velocities (Eq. 4) retrieved from successfully constrained flow vectors. Note that this is a somewhat arbitrary choice of the intrinsic uncertainty of the Farnebäck algorithm, solely based on the findings of Menze and Geiger (2015). However, we remark again, that it is beyond the scope of this paper to verify the accuracy of the Farnebäck algorithm, which we use to illustrate the performance of our new post analysis method. For the unphysical motion vectors (which are replaced by the PDV) we assume a conservative uncertainty based on the width  $n\sigma$  of the histogram peaks (cf. Appendices B3 and B5.1)."

P12, L17 - Suggest rewording this sentence for clarity: '. . . all pixels on I while Xpixok corresponds to the SO2 ICA considering only pixels showing reliable flow vectors.

We thank the reviewer for this suggestion and changed the phrasing accordingly.

P12, L21 – Is there a specific reason for performing the velocity analysis on the onband images rather than on the AA images?

Yes, the OF algorithm performed best (on visual inspection) for the on-band images. We added this information in the text:

**Changes to the manuscript**

"... on-band OD ( $\tau_{on}$ ) images, since the OF algorithm showed best performance for the onband OD images (based on visual inspection before the analysis)."

Figures 4 and 8– It's very difficult to see the vectors. Can the color scheme be improved, or the vectors made longer/bolder?

We improved the figure by extending the length of the displayed flow vectors and also, by increasing the linewidth.

**Further changes to the manuscript**

Caption of Fig. 4 now reads as (Screenshot):

Figure 4. Left: Example flow vector field (blue lines with red dots) of the Farnebäck optical flow algorithm for the Etna plume at 07:13 UTC including the two PCS lines (blue / orange) and the corresponding ROIs used for the histogram analysis (semi-transparent rectangles). Middle, right: histograms of orientation angles  $\mathcal{M}_{\varphi}$  and vector magnitudes  $\mathcal{M}_{|f|}$  for both lines (bar plot), determined using condition S7 in Appendix A1. The  $\mathcal{M}_{\varphi}$  histogram (middle) also includes fit results of the Multi-Gauss peak detection (thick solid lines). The retrieved histogram parameters ( $\varphi_{\mu}$ ,  $|f|_{\mu}$ ) and expectation intervals  $\mathcal{M}_{\varphi}$ ,  $\mathcal{M}_{|f|}$  are indicated with solid and dashed vertical lines, respectively. From the corrected *DVF*, average effective velocities of  $v_{\text{eff}} = (3.9 \pm 0.5) \text{ m/s}$  (orange line) and  $v_{\text{eff}} = (4.4 \pm 0.8) \text{ m/s}$  (blue line) were retrieved. Note that in the left image 1) vectors shorter than 1.5 pixels are excluded, 2) the displayed vector lengths were extended by a factor of 3 and 3) only every 15th pixel of the *DVF* is displayed.

**Caption of Fig. 8 now reads as (Screenshot):**

Figure 8. Left: Example output of the Farnebäck optical flow algorithm for the Guallatiri emissions at 14:48 UTC including the two example PCS lines (blue / orange line) and the corresponding ROIs. Middle and right: Histograms of magnitudes  $\mathcal{M}_{\varphi}$  and orientation angles  $\mathcal{M}_{|f|}$  used to retrieve the expectation intervals  $\mathcal{I}_{\varphi}$  and  $\mathcal{I}_{|f|}$  and the corresponding *PDV* in each ROI, respectively (cf. Eq. 5). From the latter, effective velocities of  $\mathbf{v}_{\text{eff}} = (3.1 \pm 0.5) \text{ m/s}$  (crater, orange) and  $\mathbf{v}_{\text{eff}} = (1.8 \pm 0.6) \text{ m/s}$  (fumaroles, blue) were retrieved. Note that in the left image 1) vectors shorter than 1.5 pixels are excluded, 2) the displayed vector lengths were extended by a factor of 2 and 3) only every 15th pixel of the *DVF* is displayed.

**P17, L12 – I assume that the 'flow\_histo' method should be the 'flow\_hybrid' method.**

This is true and was corrected for accordingly.

P18, L3 – '. . . gradual decrease in the effective gas velocities after 14:54 at the main crater, which could. . . I wonder whether the velo: glob method should even be discussed here if it failed to retrieve velocities for the main crater. Perhaps omit it in Figure 10 (left) and only discuss it for the fumaroles?

We agree that it is rather misleading to assume the same global velocity for the central crater emissions, which we retrieved from the fumarolic emissions, especially due to the considerably large deviations. Therefore, as suggested, we removed "velo\_glob" from the central crater results (both in Fig. 10 and Fig. 11) and updated the text and figure captions as follows:

**Changes to the manuscript Fig. 10 (Screenshot), new caption:**

Figure 10. Guallatiri SO2 emission-rates from the summit crater (left, orange colours) and the fumarolic field (right, blue colours) using the two PCS lines shown in Fig. 2. Uncertainties (shaded areas) are only plotted for the *flow\_hybrid* and the *glob* (cross-correlation based velocity) retrieval methods. Further included are the corresponding effective velocities (from the *flow\_hybrid* method) and the OF quality factors  $\kappa$  (Eq. 7). The central crater emissions show only little variability ( $\Phi \approx 0.6 \text{ kg/s}$ ) while the fumarolic emissions are characterised by a comparatively strong emission "event" at 14:55 UTC showing peak emissions of 2.5 kg/s.

**Figure 11, new caption:**

**Figure 11.** Relative deviations of Guallatiri emission-rates shown in Fig. 10. The deviations are plotted as ratios normalised to the results from the proposed method (*flow\_hybrid*), both for the crater (top, no cross-correlation results available) and for the fumarolic emissions (bottom). The average ratios are  $1.23 \pm 0.32\%$  (fumaroles, *glob*) and  $0.85 \pm 0.12$  (crater, *flow\_raw*) and  $0.75 \pm 0.22\%$  (fumaroles, *flow\_raw*). Again, the latter show a rather strong variability between the images.

**P.17, L7 – end of Sect. 3.2, updated text which now reads as:**

"The results of the emission-rate analysis are shown in Fig. 10, again, including effective velocities and  $\kappa$  values for both PCS lines (cf. Fig. 6). As in the Etna example, the SO2 emission-rates were calculated using the three different velocity retrieval methods introduced above (i.e. glob, flow\_raw, flow\_hybrid). In general, similar trends can be observed. The uncorrected OF often causes significant underestimations in the SO2 emission-rates. It furthermore accompanies rather strong (and unphysical) high-frequency fluctuations which are propagated to the SO2 emission-rates (see Sect. 3.1 for a discussion). The cross-correlation velocity analysis could only successfully be applied to the emissions from the fumarolic field (cf. Fig. 2 and Sect. B4), since the central crater plume showed too strong fluctuations both in space and time. The corresponding emission-rates of the fumarole emissions (Fig. 10, right, purple colours) show good agreement with the flow\_hybrid method.

The SO2 emission-rates, which were calculated based on the proposed flow\_hybrid method, show only little variation in the central crater emissions with values ranging between 0.1-1.5 kg/s (Fig. 10, left). The corresponding fumarole emissions, however, show rather strong variations with peak emission-rates of 2.5 kg/s (at 14:55 UTC), even exceeding the observed central crater amounts. The sum of both sources yields total SO2 emission-rates of  $\Phi_{tot} = 1.3 \pm 0.5$  kg/s with peak emissions of up to 2.9 kg/s.

Relative deviations of the retrieved SO2 emission-rates between the three velocity methods are shown in Fig. 11. As in the case of Etna, the cross-correlation based results (glob,

fumaroles) tend to be slightly increased (here: +23%) while the uncorrected OF (flow\_raw) results in an average underestimation of -20%."

Figure 11 – Similarly, it might be best to only plot the glob method for the fumaroles, not for the main crater.

We followed this suggestion (cf. previous point).

P19, L15 – '... during the observation PERIOD.'

We followed this suggestion.

P20, L3 – Etna's (please add apostrophe)

We followed this suggestion.

P20, L6 - In the case of Guallatiri, these are the first SO2 emission rates reported in the literature.

We followed this suggestion.

Changes to the manuscript P20, L6: Changed "In the case of Guallatiri, no reports are available yet related to its SO2 emissions." to "In the case of Guallatiri, these are the first SO2 emission rates reported in the literature."

Figures 1, 2, and 13 – The unit for column density is molecules/cm2. Adding the 'molecules' in plot labels would improve clarity.

We followed this suggestion and updated the y-axis labels in all three figures respectively.

Figures 14 and 15 – Is there a reason why the AA is plotted in figure 14, but only the onband optical depth is plotted in figure 15? Perhaps it's best to be consistent?

See discussion above: in the case of Guallatiri, the time-series of on-band ODs yielded better results due to artefacts in the AA arising from the temporal shift between on and off-band at the rather pronounced turbulent conditions present during the measurement.

**Review 1 (Anonymous Referee #3)**

Received and published: 26 October 2017

**General comments:**

This paper introduces a new post-correction method for the output of the Farneback optical flow algorithm. The Farneback algorithm tends to fail in low contrast regions, such as the central plume region, and apparently does not give any uncertainty estimate nor quality flags for its output that would allow a straight-forward filtering of unreliable motion vectors. The unphysical output is characterized by short randomly oriented vectors. The proposed post-correction method exploits this feature and replaces the unphysical output within a selected plume region by a local average obtained from a histogram analysis. Better velocity estimates for the plume central regions are thus obtained and the corresponding estimates of sulfur dioxide emission rates improved. Evidence on the performance of the proposed post-correction algorithm is given for two different volcano cases: Mt. Etna in Italy and Guallatiri volcano in Chile. The algorithm has many tunable parameters but appears to perform well with the selected parameters for the two cases. The results are comparable to two reference methods: a crosscorrelation method and using the uncorrected output from the Farneback method. The post-corrected emission rates are larger than the uncorrected ones, as expected when randomly oriented vectors are replaced by the local average. For the remotely located Guallatiri, estimates of the sulfur dioxide emission rate are given for the first time. The manuscript is within the aims and scope of AMT and is suitable for publication with the corrections listed below. The scientific methods are commonly used in the field and presentation quality is generally good.

**Specific comments:**

P7, L8-11: Could you please add more details on the determination of the background intensities. Do you fit functions to the image background and then evaluate the fitted functions at the plume location to get the background intensity?

We agree that this is useful information and therefore added a sentence to the manuscript which specifies the corresponding background retrieval methods of the used Pyplis software. Basically the SRI and corresponding plume images are compared in suitable sky reference areas of the plume image by fitting 1d polynomials both in the horizontal and vertical direction and based on that performing a column and row-wise correction of the original SRI. Pyplis provides different default methods that depend on the type of sky reference area used (e.g. continuous using horizontal or vertical profile lines, discrete using rectangular image areas) and the corresponding polynomial order used for the correction (i.e. linear for discrete rectangles, higher order for profile lines).

Please see the Pyplis paper https://www.preprints.org/manuscript/201710.0085/v1 for details. Please follow this link

http://pyplis.readthedocs.io/en/latest/api.html#pyplis.plumebackground.PlumeBackground Model

to see details about the implementation of the different background retrieval methods.

**Changes to the manuscript**

P.7, L.11f, added sentence: "The background retrieval was done using the background modelling methods 6 (Etna) and 4 (Guallatiri) of the used analysis software Pyplis (Gliß et al., 2017, cf. Table 2 therein)."

P10, Fig.3: Filtering out the vectors shorter than 1.5 pixels seems to effectively remove the unphysical vectors in the background. Considering the vectors in the remaining plume (red / blue shaded areas) how do you distinguish vectors due to real turbulent motion from the unphysical output of the Farneback algorithm?

In fact, it is not possible to distinguish such potential real turbulent motion vectors that may occur in a turbulent plume, from the noise vectors. However, if the plume conditions are very turbulent, the distributions of both histogram become broader due to the increased scatter in the vector parameters. The confidence interval (specified by parameter *n*) for the acceptance range of both histograms (set by the user, in this study n = 3 is used) can also be used to control, which vectors are considered noise / trash and which not. It is certainly true, that there might be *real* outliers in the plume which are not falling into the expectation intervals of the histograms at a specific confidence interval  $n\sigma$ . Such vectors would hence, be misinterpreted as ill-constrained by the histogram analysis. However, such highly local fluctuations of a *few* vectors are likely not resolved by the OF algorithm in the first place, due to the applied spatial coherency constraints, that are imposed to the OF retrieval and that are intrinsic to all OF algorithms (see first point of reviewer 1 and corresponding extended discussion on OF algorithms in the introduction of the manuscript). If it is significantly more than a *few* vectors, it will manifest in the histogram distributions and hence, be considered by the analysis.

P12, L17: Please write out the acronym ICA as it is introduced here for the first time.

We followed this suggestion.

**Changes to the manuscript**

We replaced "ICA" with "integrated-column-amount (ICA)".

P15, Fig. 7 and P19, Fig. 11: y-axis labels, legends and caption texts are inconsistent in describing what is plotted: absolute deviation  $\Delta \Phi$ , relative deviation  $\Delta \Phi / \Phi_{hybrid}$  or ratio  $\Phi / \Phi_{hybrid}$ . The legend appears to be correct, i.e. ratios are plotted. Please make these consistent.

We followed this suggestion.

**Changes to the manuscript**

We updated y-labels and legends appropriately in Figs. 7 and 11.

P19, L5: There is no evidence in the paper to make a generic statement about OF algorithms. Only the Farneback algorithm was applied to two cases. Please be more specific, for example: "We showed that the Farneback algorithm is able to resolve ..."

We followed this suggestion.

**Changes to the manuscript**

We replaced "We showed, that OF algorithms are (generally) able to resolve ..." with "We showed that the Farnebäck OF algorithm is (generally) able to resolve ..."

P25, Table 2: How sensitive is your method to the selection of parameters for the Farneback algorithm and the histogram analysis given here, and also for the selection of the ROI? Did you perform any sensitivity analysis? Is it just trial and error to arrive at the given values for the parameters? Why are both  $\tau_{min}$  and  $|f|_{min}$  needed in the histogram analysis, as Fig. 3 suggests that  $|f|_{min}$  alone selects the plume region?

We agree with the reviewer that the sensitivity of our new *post* analysis method is an important point and we extended Appendix by a sensitivity study, which investigates the percentage impact of the choice of the length threshold |f|min and the chosen confidence level n (for the acceptance intervals of the peak widths) on the retrieved emission-rates (for details see "Changes to the manuscript" below). However, we wish to remark again, that our method is not specific to the Farneback OF algorithm, which we use to illustrate, that the proposed method can significantly improve the robustness of the results. Namely, that the errors and uncertainties are significantly smaller and easier to assess, if this post analysis is used (compared to the raw case which does not provide any measurable information related to the performance of the OF algorithm). Even though there might be other OF algorithms that come with a confidence measure we remark, that the choice of a suitable OF algorithm is often a trade-off between computational efficiency and performance and further, users often have to rely on existing code / implementations of literature OF algorithms (as pointed out in the new Introduction, see review 1, first point). The strength of our method is, that indeed, it is independent of the choice of the OF algorithm and as we show, its computational demands do not state a significant increase in the emission-rate analysis since the OF computation itself is a typically very demanding operation (actually, looking at the results from the provided OF benchmarks in the paper, the Farneback can be considered a fast OF algorithm). For these very reasons (i.e. that this study is not about investigating the Farneback algorithm), we did not perform a systematic sensitivity test of the impact of the Farneback algorithm parameters (e.g. number of iterations, Gaussian window width, parameters for polynomial expansion) on the retrieved emission rates, since this is beyond the scope of this paper and somewhat misleading, since the focus is on the post analysis method. However, we remark that we did many qualitative tests regarding both the OF performance itself in dependency of the Farneback input parameters and that we also qualitatively investigated sensitivity to the other parameters of the histogram analysis (e.g.  $\tau_{min}$ , cf. Table 2 in Appendix) and found that these impacts are small compared to the two most important ones (for typical / reasonable values of each of the parameters).

**Changes to the manuscript**

Added subsection B.5.1 "Sensitivity to the chosen histogram analysis settings" at the end of the Appendix (in Sect. B5 "SO2 emission-rate uncertainties"):

**"B5.1 Sensitivity to the chosen histogram analysis settings**

The sensitivity of the retrieved emission-rates (cf. Sect. 3) to the input parameters  $|f|_{min}$  and  $n\sigma$  of the proposed histogram correction method (cf. Appendix B3) was investigated. These two parameters have the largest impact on the results since they determine, which flow vectors are considered ill-constrained and which ones not. The sensitivity analysis was performed using the proposed flow\_hybrid method applied to 30 images from the Etna dataset that were not corrected for the signal-dilution effect, since the latter is irrelevant for this study (all other analysis settings are the same as described in Sect. 2.3). Figure 16 shows the results of this investigation. The choice of n has a rather small impact on the emission-rates, whereas the choice of  $|f|_{min}$  impacts within a range of approximately  $\pm 17\%$ . However, considering the more realistic interval of 1-2 for  $|f|_{min}$ , the impact is less than 8%

Furthermore, we included this new Figure 16 (Screenshot):

Figure 16. Sensitivity of SO2 emission-rates as a function of the chosen input settings  $|\mathbf{f}|_{\min}$  (y-axis) and the confidence level *n* (x-axis). The investigated value ranges are 0-4 pixels for  $|\mathbf{f}|_{\min}$  and 1-3 for the *n* and the deviations are plotted as percentage deviations  $\Delta\Phi_{SO2}$  from the average SO2 emission-rate retrieved from this study. The latter amounts to 2.4 kg/s (not dilution corrected, see text). The analysis was performed using the average SO2 emission-rates retrieved from 30 images of the Etna dataset.

**Suggestions for technical improvements:**

P4, L15 and throughout the text: please do not abbreviate the word "Table" (see Manuscript preparation guidelines).

We follow this suggestion in the revised version of the manuscript.

**Changes to the manuscript**

We replaced all occurrences of "Tab." with "Table".

P6, Fig. 2: there is no label (b). Please remove the "in (b)" at the end of the caption or label the panels as (a) and (b).

We follow this suggestion in the revised version of the manuscript (see also discussion above in review 1 for further changes applied to the Figure 2 and caption).

P6, L5: replace "data is" with "data are".

We follow this suggestion in the revised version of the manuscript.

P8, L11 and throughout the text: please use "Figure" at the beginning of a new sentence, not "Fig.".

We follow this suggestion in the revised version of the manuscript.

**Changes to the manuscript**

We replaced all occurrences of "Fig." at the beginning of a new sentence with "Figure".

P17, L12: replace "flow\_histo" with "flow\_hybrid"

This issue has already been resolved in our response to reviewer 1 (see above).

P20, L3: replace "Etnas" with "Etna's"

This issue has already been resolved in our response to reviewer 1 (see above).